# Robust Federated Learning:
# The Case of Affine Distribution Shifts

**Amirhossein Reisizadeh**[*]
ECE Department
UC Santa Barbara
reisizadeh@ucsb.edu

**Farzan Farnia**[*]
LIDS
MIT
farnia@mit.edu

**Ramtin Pedarsani**
ECE Department
UC Santa Barbara
ramtin@ece.ucsb.edu

**Ali Jadbabaie**
LIDS
MIT
jadbabai@mit.edu

## Abstract

Federated learning is a distributed paradigm for training models using samples distributed across multiple users in a network, while keeping the samples on users' devices with the aim of efficiency and protecting users privacy. In such settings, the training data is often statistically heterogeneous and manifests various distribution shifts across users, which degrades the performance of the learnt model. The primary goal of this paper is to develop a robust federated learning algorithm that achieves satisfactory performance against distribution shifts in users' samples. To achieve this goal, we first consider a *structured* affine distribution shift in users' data that captures the device-dependent data heterogeneity in federated settings. This perturbation model is applicable to various federated learning problems such as image classification where the images undergo device-dependent imperfections, e.g. different intensity, contrast, and brightness. To address affine distribution shifts across users, we propose a **F**ederated **L**earning framework **R**obust to **A**ffine distribution shifts (`FLRA`) that is robust against affine distribution shifts to the distribution of observed samples. To solve the `FLRA`'s distributed minimax optimization problem, we propose a fast and efficient optimization method and provide convergence and performance guarantees via a gradient Descent Ascent (GDA) method. We further prove generalization error bounds for the learnt classifier to show proper generalization from empirical distribution of samples to the true underlying distribution. We perform several numerical experiments to empirically support `FLRA`. We show that an affine distribution shift indeed suffices to significantly decrease the performance of the learnt classifier in a new test user, and our proposed algorithm achieves a significant gain in comparison to standard federated learning and adversarial training methods.

## 1 Introduction

*Federated learning* is a new framework for training a centralized model using data samples distributed over a network of devices, while keeping data localized. Federated learning comes with the promise of training accurate models using local data points such that the privacy of participating devices is preserved; however, it faces several challenges ranging from developing statistically and computationally efficient algorithms to guaranteeing privacy.

A typical federated learning setting consists of a network of hundreds to millions of devices (nodes) which interact with each other through a central node (a parameter server). Communicating messages over such a large-scale network can lead to major slow-downs due to communication bandwidth bottlenecks [1, 2]. In fact, the communication bottleneck is one of the main grounds that distinguishes federated and standard distributed learning paradigms. To reduce communication load in federated

---

[*]Equal contribution

learning, one needs to depart from the classical setting of distributed learning in which updated local models are communicated to the central server *at each iteration*, and communicate less frequently.

Another major challenge in federated learning is the statistical heterogeneity of training data [1, 2]. As mentioned above, a federated setting involves many devices, each generating or storing personal data such as images, text messages or emails. Each user's data samples can have a (slightly) different underlying distribution which is another key distinction between federated learning and classical learning problems. Indeed, it has been shown that standard federated methods such as FedAvg [3] which are designed for i.i.d. data significantly suffer in statistical accuracy or even diverge if deployed over non-i.i.d. samples [4]. Device-dependency of local data along with privacy concerns in federated tasks does not allow learning the distribution of individual users and necessitates novel algorithmic approaches to learn a classifier robust to distribution shifts across users. Specifically, statistical heterogeneity of training samples in federated learning can be problematic for generalizing to the distribution of a test node unseen in training time. We show through various numerical experiments that even a simple linear filter applied to the test samples will suffice to significantly degrade the performance of a model learned by FedAvg in standard image recognition tasks.

To address the aforementioned challenges, we propose a new federated learning scheme called FLRA, a **F**ederated **L**earning framework with **R**obustness to **A**ffine distribution shifts. FLRA has a small communication overhead and a low computation complexity. The key insight in FLRA is model the heterogeneity of training data in a device-dependent manner, according to which the samples stored on the $i$th device $\mathbf{x}^i$ are shifted from a ground distribution by an affine transformation $\mathbf{x}^i \rightarrow \Lambda^i \mathbf{x}^i + \delta^i$. To further illustrate this point, consider a federated image classification task where each mobile device maintains a collection of images. The images taken by a camera are similarly distorted depending on the intensity, contrast, blurring, brightness and other characteristics of the camera [5, 6], while these features vary across cameras. In addition to camera imperfections, such unseen distributional shifts also originate from changes in the physical environment, e.g. weather conditions [7]. Compared to the existing literature, our model provides more robustness compared to the well-known adversarial training models $\mathbf{x}^i \rightarrow \mathbf{x}^i + \delta^i$ with solely additive perturbations [8, 9, 10], i.e. $\Lambda^i = I$. Our perturbation model also generalizes the universal adversarial training approach in which *all* the training samples are distorted with an identical perturbation $\mathbf{x}^i \rightarrow \mathbf{x}^i + \delta$ [11].

Based on the above model, FLRA formulates the robust learning task as a minimax robust optimization problem, which finds a *global* model $w^*$ that minimizes the total loss induced by the worst-case *local* affine transformations $(\Lambda^{i*}, \delta^{i*})$. One approach to solve this minimax problem is to employ techniques from adversarial training in which for each iteration and a given global model $w$, each node optimizes its own local adversarial parameters $(\Lambda^i, \delta^i)$ and a new model is obtained. This approach is however undesirable in federated settings since it requires extensive computation resources at each device as they need to fully solve the adversarial optimization problem at each iteration. To tackle this challenge, one may propose to use standard distributed learning frameworks in which each node updates its local adversarial parameters and shares with the server at *each iteration* of the distributed algorithm to obtain the updated global model. This is also in contrast with the availability of limited communication resources in federated settings. The key contribution of our work is to develop a novel method called FedRobust, which is a gradient descent ascent (GDA) algorithm to solve the minimax robust optimization problem, can be efficiently implemented in a federated setting, and comes with strong theoretical guarantees. While the FLRA minimax problem is in general non-convex non-concave, we show that FedRobust which alternates between the perturbation and parameter model variables will converge to a stationary point in the minimax objective that satisfies the Polyak-Łojasiewicz (PL) condition. Our optimization guarantees can also be extended to more general classes of non-convex non-concave distributed minimax optimization problems.

As another major contribution of the paper, we use the PAC-Bayes framework [12, 13] to prove a generalization error bound for FLRA's learnt classifier. Our generalization bound applies to multi-layer neural network classifiers and is based on the classifier's Lipschitzness and smoothness coefficients. The generalization bound together with our optimization guarantees suggest controlling the neural network classifier's complexity through Lipschitz regularization methods. Regarding FLRA's robustness properties, we connect the minimax problem in FLRA to a distributionally robust optimization problem [14, 15] where we use an optimal transport cost to measure the distance between distributions. This connection reveals that the FLRA's minimax objective provides a lower-bound for the objective of a distributionally robust problem. Finally, we discuss the results of several numerical experiments to empirically support the proposed robust federated learning method. Our experiments suggest a

significant gain under affine distribution shifts compared to existing adversarial training algorithms. In addition, we show that the trained classifier performs robustly against standard FGSM and PGD adversarial attacks, and outperforms `FedAvg`.

**Related work.** As a practical on-device learning paradigm, federated learning has recently gained significant attention in machine learning and optimization communities. Since the introduction of `FedAvg` [3] as a communication-efficient federated learning method, many works have developed federated methods under different settings with optimization guarantees for a variety of loss functions [16, 17]. Moreover, another line of work has tackled the communication bottleneck in federated learning via compression and sparsification methods [18, 19, 20]. [21, 22, 23, 24] have focused on designing privacy-preserving federated learning schemes. There have also been several recent works the study local-SGD methods as a subroutine of federated algorithms and provide various convergence results depending on the loss function class [25, 26, 27]. Making federated learning methods robust to non-i.i.d. data has also been the focus of several works [4, 28, 29].

Adversarially robust learning paradigms usually involve solving a minimax problem of the form $\min_{\boldsymbol{w}} \max_{\boldsymbol{\psi}} f(\boldsymbol{w}, \boldsymbol{\psi})$. As the theory of adversarially robust learning surges, there has been thriving recent interests in solving the minimax problem for nonconvex cases. Most recently, [30] provides nonasymptotic analysis for nonconvex-concave settings and shows that the iterates of a simple Gradient Descent Ascent (GDA) efficiently find the stationary points of the function $\Phi(\boldsymbol{w}) \coloneqq \max_{\boldsymbol{\psi}} f(\boldsymbol{w}, \boldsymbol{\psi})$. [31] establishes convergence results for the nonconvex-nonconcave setting and under PL condition. This problem has been studied in the context of game theory as well [32].

## 2   Federated Learning Scenario

Consider a federated learning setting with a network of $n$ nodes (devices) connected to a server node. We assume that for every $1 \le i \le n$ the $i$th node has access to $m$ training samples in $S^i = \{(\mathbf{x}_j^i, y_j^i) \in \mathbb{R}^d \times \mathbb{R} : 1 \le j \le m\}$. For a given loss function $\ell$ and function class $\mathcal{F} = \{f_{\boldsymbol{w}} : \boldsymbol{w} \in \mathcal{W}\}$, the classical federated learning problem is to fit the best model $\boldsymbol{w}$ to the $nm$ samples via solving the following empirical risk minimization (ERM) problem:

$$\min_{\boldsymbol{w} \in \mathcal{W}} \quad \frac{1}{nm} \sum_{i=1}^{n} \sum_{j=1}^{m} \ell\left(f_{\boldsymbol{w}}(\mathbf{x}_j^i), y_j^i\right).$$

As we discussed previously, the training data is statistically heterogeneous across the devices. To capture the non-identically-distributed nature of data in federated learning, we assume that the data points of each node have a local distribution shift from a common distribution. To be more precise, we assume that each sample stored in node $i$ in $S^i$ is distributed according to an affine transformation $h^i$ of a universal underlying distribution $P_{\mathbf{X},Y}$, i.e., transforming the features of a sample $(\mathbf{x}, y) \sim P_{\mathbf{X},Y}$ according to the following affine function $h^i(\mathbf{x}) \coloneqq \Lambda^i \mathbf{x} + \delta^i$. Here $\Lambda^i \in \mathbb{R}^{d \times d}$ and $\delta^i \in \mathbb{R}^d$, with $d$ being the dimension of input variable $\mathbf{x}$, characterize the affine transformation $h^i$ at node $i$. According to this model, all samples stored at node $i$ are affected with the same affine transformation while other nodes $j \ne i$ may experience different transformations.

This *structured* model particularly supports the data heterogeneity in federated settings. That is, the data generated and stored in each federated device is exposed to identical yet device-dependent distortions while different devices undergo different distortions. As an applicable example that manifests the proposed perturbation model, consider a federated image classification task over the images taken and maintained by mobile phone devices. Depending on the environment's physical conditions and the camera's imperfections, the pictures taken by a particular camera undergo device-dependent perturbations. According to the proposed model, such distribution shift is captured as an affine transformation $h^i(\mathbf{x}) = \Lambda^i \mathbf{x} + \delta^i$ on the samples maintained by node $i$. To control the perturbation power, we consider bounded Frobenius and Euclidean norms $\|\Lambda - I_d\|_F \le \epsilon_1$ and $\|\delta\|_2 \le \epsilon_2$ enforcing the affine transformation to have a bounded distance from the identity transformation.

Based on the model described above, our goal is to solve the following distributionally robust federated learning problem:

$$\min_{\boldsymbol{w} \in \mathcal{W}} \quad \frac{1}{n} \sum_{i=1}^{n} \max_{\substack{\|\Lambda^i - I\|_F \le \epsilon_1 \\ \|\delta^i\| \le \epsilon_2}} \quad \frac{1}{m} \sum_{j=1}^{m} \ell\left(f_{\boldsymbol{w}}(\Lambda^i \mathbf{x}_j^i + \delta^i), y_j^i\right). \tag{1}$$

The minimax problem (1) can be interpreted as $n+1$ coupled optimization problems. First, in $n$ inner local maximization problems and for a given global model $\boldsymbol{w}$, each node $1 \le i \le n$ seeks a (feasible) affine transformation $(\Lambda^i, \delta^i)$ which results in high losses via solving $\max_{\Lambda^i, \delta^i} \frac{1}{m} \sum_{j=1}^m \ell(f_{\boldsymbol{w}}(\Lambda^i \mathbf{x}_j^i + \delta^i), y_j^i)$ over its $m$ training samples in $S^i$. Then, the outer minimization problem finds a global model yielding the smallest value of cumulative losses over the $n$ nodes.

Solving the above minimax problem requires collaboration of distributed nodes via the central server. In federated learning paradigms however, such nodes are entitled to limited computation and communication resources. Such challenges particularly prevent us from employing the standard techniques in adversarial training and distributed ERM. More precisely, each iteration of adversarial training requires solving a maximization problem at each local node which incurs extensive computational cost. On the other hand, tackling the minimax problem (1) via iterations of standard distributed learning demands frequent message-passing between the nodes and central server *at each iteration*, hence yielding massive communication load on the network. To account for such system challenges, we constitute our goal to solve the robust minimax problem in (1) with small computation and communication cost so that it can be feasibly and efficiently implemented in a federated setting.

## 3   The Proposed `FedRobust` Algorithm

To guard against affine distribution shifts, we propose to change the original constrained maximization problem to the following worst-case loss at each node $i$, given a Lagrange multiplier $\lambda > 0$:

$$\max_{\Lambda^i, \delta^i} \frac{1}{m} \sum_{j=1}^m \ell\left(f_{\boldsymbol{w}}(\Lambda^i \mathbf{x}_j^i + \delta^i), y_j^i\right) - \lambda \|\Lambda^i - I\|_F^2 - \lambda \|\delta^i\|_2^2. \tag{2}$$

Here we use a norm-squared penalty requiring a bounded distance between the feasible affine transformations and the identity mapping, and find the worst-case affine transformation that results in the maximum loss for the samples of node $i$. By averaging such worst-case local losses over all the $n$ nodes and minimizing w.r.t. model $\boldsymbol{w}$, we reach the following minimax optimization problem:

$$\min_{\boldsymbol{w} \in \mathcal{W}} \max_{(\Lambda^i, \delta^i)_{i=1}^n} \frac{1}{nm} \sum_{i=1}^n \sum_{j=1}^m \ell\left(f_{\boldsymbol{w}}(\Lambda^i \mathbf{x}_j^i + \delta^i), y_j^i\right) - \lambda \|\Lambda^i - I\|_F^2 - \lambda \|\delta^i\|_2^2. \tag{3}$$

This formalizes our approach to tackling the robust federated learning problem, which we call "**F**ederated **L**earning framework **R**obust to **A**ffine distribution shift" or `FLRA` in short.

In order to solve `FLRA` in (3), we propose a gradient optimization method that is computationally and communication-wise efficient, called `FedRobust`. The proposed `FedRobust` algorithm is an iterative scheme that applies stochastic gradient descent ascent (SGDA) updates for solving the minimax problem (3). As summarized in Algorithm 1, in each iteration $t$ of local updates, each node $i$ takes a (stochastic) gradient ascent step and updates its affine transformation parameters $(\Lambda_t^i, \delta_t^i)$. It also updates the local classifier's parameters $\boldsymbol{w}_t^i$ via a gradient descent step. After $\tau$ local iterations, local models $\boldsymbol{w}_t^i$ are uploaded to the server node where the global model is obtained by averaging the local ones. The averaged model is then sent back to the nodes to begin the next round of local iterations with this fresh initialization. Note that each node updates its perturbation parameters only once in each iteration which yields light computation cost as opposed to standard adversarial training methods. Moreover, periodic communication at every $\tau$ iterations, reduces the communication load compared to standard distributed optimization methods by a factor $\tau$.

---

**Algorithm 1** `FedRobust`

    **Input:** $\{\boldsymbol{w}_0^i = \boldsymbol{w}_0, \Lambda_0^i, \delta_0^i\}_{i=1}^n, \eta_1, \eta_2, \tau, T$
1: **for** each itr. $0 \le t \le T-1$, node $i$ computes

$$\Lambda_{t+1}^i = \Lambda_t^i + \eta_2 \tilde{\nabla}_\Lambda f^i(\boldsymbol{w}_t^i, \Lambda_t^i, \delta_t^i)$$
$$\delta_{t+1}^i = \delta_t^i + \eta_2 \tilde{\nabla}_\delta f^i(\boldsymbol{w}_t^i, \Lambda_t^i, \delta_t^i)$$

2:    **if** $t$ does not divide $\tau$ **then**

$$\boldsymbol{w}_{t+1}^i = \boldsymbol{w}_t^i - \eta_1 \tilde{\nabla}_{\boldsymbol{w}} f^i(\boldsymbol{w}_t^i, \Lambda_t^i, \delta_t^i)$$

3:    **else** node $i$ uploads to server:
        $\boldsymbol{w}_t^i - \eta_1 \tilde{\nabla}_{\boldsymbol{w}} f^i(\boldsymbol{w}_t^i, \Lambda_t^i, \delta_t^i)$
4:    server sends to all nodes $i$:

$$\boldsymbol{w}_{t+1}^i = \frac{1}{n} \sum_{j=1}^n \left[\boldsymbol{w}_t^j - \eta_1 \tilde{\nabla}_{\boldsymbol{w}} f^j(\boldsymbol{w}_t^j, \Lambda_t^j, \delta_t^j)\right]$$

5:    **end if**
6: **end for**
    **Output:** $\overline{\boldsymbol{w}}_T = \frac{1}{n} \sum_{i=1}^n \boldsymbol{w}_T^i$

---

It is worth noting that the local affine transformation variables $\Lambda^i, \delta^i$ are coupled even though they remain on their corresponding nodes and are not exchanged with the server. This is due to the fact

that the fresh model $\boldsymbol{w}$ is the average of the updated models from *all* the nodes; hence, updating $\Lambda^i, \delta^i$ for node $i$ will affect $\Lambda^j, \delta^j$ for other nodes $j \neq i$ in the following iterations. This is indeed a technical challenge that arises in proving the optimization guarantees of `FedRobust` in Section 4.1.

# 4 Theoretical Guarantees: Optimization, Generalization and Robustness

In this section, we establish our main theoretical results. First, we characterize the convergence of `FedRobust` in Algorithm 1. Next, we prove that the learned hypothesis will properly generalize from training data to unseen test samples. Lastly, we demonstrate that solving the `FLRA`'s minimax problem (3) results in a robust classifier to Wasserstein shifts structured across the nodes.

## 4.1 Optimization guarantees

In this section, we establish our main convergence results and show that `FedRobust` finds saddle points of the minimax problem in (2) for two classes of loss functions. We first set a few notations as follows. We let matrix $\boldsymbol{\psi}^i = (\Lambda^i, \delta^i) \in \mathbb{R}^{d \times (d+1)}$ denote the joint transformation variables corresponding to node $i$. The collection of $n$ such variables corresponding to the $n$ nodes is denoted by the matrix $\Psi = (\boldsymbol{\psi}^1; \cdots; \boldsymbol{\psi}^n)$. We can now rewrite the minimax problem (3) as follows:

$$\min_{\boldsymbol{w}} \max_{\Psi} f(\boldsymbol{w}, \Psi) \coloneqq \min_{\boldsymbol{w}} \max_{\boldsymbol{\psi}^1, \cdots, \boldsymbol{\psi}^n} \frac{1}{n} \sum_{i=1}^{n} f^i(\boldsymbol{w}, \boldsymbol{\psi}^i), \tag{4}$$

where $f$ and $f^i$s denote the penalized global and local losses, respectively; that is, for each node $i$

$$f^i(\boldsymbol{w}, \boldsymbol{\psi}^i) \coloneqq \frac{1}{m} \sum_{j=1}^{m} \ell\left(f_{\boldsymbol{w}}(\Lambda^i \mathbf{x}_j^i + \delta^i), y_j^i\right) - \lambda \|\Lambda^i - I\|_F^2 - \lambda \|\delta^i\|^2. \tag{5}$$

We also define $\Phi(\boldsymbol{w}) \coloneqq \max_{\Psi} f(\boldsymbol{w}, \Psi)$ and $\Phi^* \coloneqq \min_{\boldsymbol{w}} \Phi(\boldsymbol{w})$. Next, we state a few customary assumptions on the data and loss functions. As we mentioned before, we assume that data is heterogeneous (non-iid). There are several notions to quantify the degree of heterogeneity in the data. In this work we use a notion called *non-iid degree* which is defined as the variance of the local gradients with respect to a global gradient [33].

**Assumption 1** (Bounded non-iid degree). *We assume that when there are no perturbations, the variance of the local gradients with respect to the global gradient is bounded. That is, there exists $\rho_f^2$ such that*

$$\frac{1}{n} \sum_{i=1}^{n} \left\|\nabla_{\boldsymbol{w}} f^i(\boldsymbol{w}, \boldsymbol{\psi}^i) - \nabla_{\boldsymbol{w}} f(\boldsymbol{w}, \Psi)\right\|^2 \leq \rho_f^2, \quad \text{for } \boldsymbol{\psi}^i = (I, 0), \Psi = (\boldsymbol{\psi}^1; \cdots; \boldsymbol{\psi}^n), \text{ and } \forall \boldsymbol{w}.$$

**Assumption 2** (Stochastic gradients). *For each node $i$, the stochastic gradients $\tilde{\nabla}_{\boldsymbol{w}} f^i$ and $\tilde{\nabla}_{\boldsymbol{\psi}} f^i$ are unbiased and have variances bounded by $\sigma_{\boldsymbol{w}}^2$ and $\sigma_{\boldsymbol{\psi}}^2$, respectively. That is,*

$$\mathbb{E}\left\|\tilde{\nabla}_{\boldsymbol{w}} f^i(\boldsymbol{w}, \boldsymbol{\psi}) - \nabla_{\boldsymbol{w}} f^i(\boldsymbol{w}, \boldsymbol{\psi})\right\|^2 \leq \sigma_{\boldsymbol{w}}^2, \quad \mathbb{E}\left\|\tilde{\nabla}_{\boldsymbol{\psi}} f^i(\boldsymbol{w}, \boldsymbol{\psi}) - \nabla_{\boldsymbol{\psi}} f^i(\boldsymbol{w}, \boldsymbol{\psi})\right\|^2 \leq \sigma_{\boldsymbol{\psi}}^2, \quad \forall \boldsymbol{w}, \boldsymbol{\psi}.$$

**Assumption 3** (Lipschitz gradients). *All local loss functions have Lipschitz gradients. That is, for any node $i$, there exist constants $L_1, L_2, L_{12},$ and $L_{21}$ such that for any $\boldsymbol{w}, \boldsymbol{w}', \boldsymbol{\psi}, \boldsymbol{\psi}'$ we have*

$$\left\|\nabla_{\boldsymbol{w}} f^i(\boldsymbol{w}, \boldsymbol{\psi}) - \nabla_{\boldsymbol{w}} f^i(\boldsymbol{w}', \boldsymbol{\psi})\right\| \leq L_1 \left\|\boldsymbol{w} - \boldsymbol{w}'\right\|, \quad \left\|\nabla_{\boldsymbol{w}} f^i(\boldsymbol{w}, \boldsymbol{\psi}) - \nabla_{\boldsymbol{w}} f^i(\boldsymbol{w}, \boldsymbol{\psi}')\right\| \leq L_{12} \left\|\boldsymbol{\psi} - \boldsymbol{\psi}'\right\|_F,$$

$$\left\|\nabla_{\boldsymbol{\psi}} f^i(\boldsymbol{w}, \boldsymbol{\psi}) - \nabla_{\boldsymbol{\psi}} f^i(\boldsymbol{w}', \boldsymbol{\psi})\right\|_F \leq L_{21} \left\|\boldsymbol{w} - \boldsymbol{w}'\right\|, \quad \left\|\nabla_{\boldsymbol{\psi}} f^i(\boldsymbol{w}, \boldsymbol{\psi}) - \nabla_{\boldsymbol{\psi}} f^i(\boldsymbol{w}, \boldsymbol{\psi}')\right\|_F \leq L_2 \left\|\boldsymbol{\psi} - \boldsymbol{\psi}'\right\|_F.$$

We show the convergence of `FedRobust` for two classes of loss functions: PL-PL and nonconvex-PL. Next, we briefly describe these classes and state the main results. The celebrated work of Polyak [34] introduces a sufficient condition for an unconstrained minimization problem $\min_x g(x)$ under which linear convergence rates can be established using gradient methods. A function $g(x)$ satisfies the Polyak-Łojasiewicz (PL) condition if $g^* = \min_x g(x)$ exits and is bounded, and there exists a constant $\mu > 0$ such that $\|\nabla g(x)\|^2 \geq 2\mu(g(x) - g^*), \forall x$. Similarly, we can define two-sided PL condition for our minimax objective function in (4) [31].

**Assumption 4** (PL condition). *The global function $f$ satisfies the two-sided PL condition, that is, there exist positive constants $\mu_1$ and $\mu_2$ such that*

(i) $\dfrac{1}{2\mu_1} \left\|\nabla_{\boldsymbol{w}} f(\boldsymbol{w}, \Psi)\right\|^2 \geq f(\boldsymbol{w}, \Psi) - \min_{\boldsymbol{w}} f(\boldsymbol{w}, \Psi)$, (ii) $\dfrac{1}{2\mu_2} \left\|\nabla_{\Psi} f(\boldsymbol{w}, \Psi)\right\|_F^2 \geq \max_{\Psi} f(\boldsymbol{w}, \Psi) - f(\boldsymbol{w}, \Psi)$.

In other words, Assumptions 4 states that the functions $f(\cdot, \Psi)$ and $-f(\boldsymbol{w}, \cdot)$ satisfy the PL condition with constants, $\mu_1$ and $\mu_2$, respectively. To measure the optimality gap at iteration $t$, we define the potential function $P_t \coloneqq a_t + \beta b_t$, where $a_t \coloneqq \mathbb{E}[\Phi(\overline{\boldsymbol{w}}_t)] - \Phi^*$ and $b_t \coloneqq \mathbb{E}[\Phi(\overline{\boldsymbol{w}}_t) - f(\overline{\boldsymbol{w}}_t, \Psi_t)]$ and $\beta$ is an arbitrary and positive constant. Note that both $a_t$ and $b_t$ are non-negative and if $P_t$ approaches zero, it implies that $(\overline{\boldsymbol{w}}_t, \Psi_t)$ is approaching a minimax point.

**Theorem 1** (PL-PL loss). *Consider the iterates of* FedRobust *in Algorithm 1 and let Assumptions 1, 3, and 4 hold. Then for any iteration $t \geq 0$, the optimality gap $P_t \coloneqq a_t + \frac{1}{2}b_t$ satisfies the following:*

$$P_t \leq \left(1 - \frac{1}{2}\mu_1\eta_1\right)^t P_0 + 32\eta_1\frac{\tilde{L}}{\mu_1}(\tau-1)^2\rho^2 + 8\eta_1\frac{\tilde{L}}{\mu_1}(\tau-1)(n+1)\frac{\sigma_{\boldsymbol{w}}^2}{n} + \eta_1\frac{\hat{L}}{\mu_1}\frac{\sigma_{\boldsymbol{w}}^2}{n} + \frac{\eta_2^2}{\eta_1}\frac{L_2}{2\mu_1}\sigma_\psi^2,$$

*for maximization step-size $\eta_2$ and minimization step-size $\eta_1$ that satisfy the following conditions:*

$$\eta_2 \leq \frac{1}{L_2}, \quad 32\eta_1^2(\tau-1)^2 L_1^2 \leq 1, \quad \frac{\mu_2^2\eta_2 n}{\eta_1 L_1 L_2} \geq 1 + 8\frac{L_{12}^2}{L_1 L_2}, \quad \eta_1\left(\hat{L} + \frac{80\tilde{L}(\tau-1)}{\mu_1\eta_1(1-\frac{1}{2}\mu_1\eta_1)^{\tau-1}}\right) \leq 1.$$

*Here, we denote $\rho^2 \coloneqq 3\rho_f^2 + 6L_{12}^2(\epsilon_1^2 + \epsilon_2^2)$ where $\epsilon_1$ and $\epsilon_2$ specify the bounds on the affine transformations $h^i(\mathbf{x}) = \Lambda^i\mathbf{x} + \delta^i$. We also use the following notations:*

$$L_\Phi = L_1 + \frac{L_{12}L_{21}}{2n\mu_2}, \quad \tilde{L} = \frac{3}{2}\eta_1 L_1^2 + \frac{1}{2}\eta_2 L_{21}^2, \quad \hat{L} = \frac{3}{2}L_\Phi + \frac{1}{2}L_1 + \frac{L_{21}^2}{L_2}.$$

Special cases of this convergence result is consistent with similar ones already established in the literature. In the particular case of (non-federated) distributed optimization, i.e. $\tau = 1$, Theorem 1 recovers the convergence result in [31]. Moreover, putting $\epsilon_1, \epsilon_2 \to 0$ reduces the problem to standard (non-robust) federated learning where our result is also consistent with the prior work [16]. We also note that the conditions on the stepsizes can be interpreted as linear conditions on $\eta_1, \eta_2$ and is always feasible. For instance, one can pick $\eta_1 = \mathcal{O}(\ln(T)/T), \eta_2 = \mathcal{O}(\ln(T)/T)$ for running FedRobust for $T$ iterations, which yields that $P_T \leq \mathcal{O}(\ln(T)/T)$. Next, we relax the PL condition on $f(\cdot, \Psi)$ stated in Assumption 4 (i) and show that the iterates of the FedRobust method find a stationary point of the minimax problem (4) when the objective function $f(\boldsymbol{w}, \Psi)$ only satisfies the PL condition with respect to $\Psi$ and is nonconvex with respect to $\boldsymbol{w}$.

**Theorem 2** (Nonconvex-PL loss). *Consider the iterates of* FedRobust *in Algorithm 1 and let Assumptions 1, 3, and 4 (ii) hold. Then, the iterates of* FedRobust *after $T$ iterations satisfy:*

$$\frac{1}{T}\sum_{t=0}^{T-1}\mathbb{E}\left\|\nabla\Phi(\overline{\boldsymbol{w}}_t)\right\|^2 \leq \frac{4\Delta_\Phi}{\eta_1 T} + \frac{4L_2^2}{\mu_2^2 n^2}\frac{\epsilon^2}{\eta_1 T} + 64\eta_1\tilde{L}(\tau-1)^2\rho^2 + 16\eta_1\tilde{L}(\tau-1)\frac{n+1}{n}\sigma_{\boldsymbol{w}}^2 + 2\eta_1\hat{L}\frac{\sigma_{\boldsymbol{w}}^2}{n} + \frac{\eta_2^2}{\eta_1}L_2\sigma_\psi^2,$$

*with $\tilde{L}, \hat{L}, L_\Phi, \rho^2$ defined in Theorem 1, $\epsilon^2 \coloneqq \epsilon_1^2 + \epsilon_2^2$ and $\Delta_\Phi \coloneqq \Phi(\boldsymbol{w}_0) - \Phi^*$, if step-sizes $\eta_1, \eta_2$ satisfy*

$$\eta_2 \leq \frac{1}{L_2}, \quad \frac{\eta_1}{\eta_2} \leq \frac{\mu_2^2 n^2}{8L_{12}^2}, \quad 32\eta_1^2(\tau-1)^2 L_1^2 \leq 1, \quad \eta_1\left(\hat{L} + 40\tilde{L}(\tau-1)^2\right) \leq 1.$$

It is worth noting this theorem also recovers the existing results for distributed minimax optimization, i.e. $\tau = 1$ [30] and standard federated learning for nonconvex objectives, i.e. $\epsilon_1, \epsilon_2 \to 0$ [20, 27].

### 4.2 Generalization guarantees

Following the margin-based generalization bounds developed in [13, 35, 36], we consider the following margin-based error measure for analyzing the generalization error in FLRA with general neural network classifiers:

$$\mathcal{L}_\gamma^{\mathrm{adv}}(\boldsymbol{w}) \coloneqq \frac{1}{n}\sum_{i=1}^n \Pr_i\left(f_{\boldsymbol{w}}(h_{adv}^i(\mathbf{X}))[Y] - \max_{j \neq Y} f_{\boldsymbol{w}}(h_{adv}^i(\mathbf{X}))[j] \leq \gamma\right). \tag{6}$$

Here, $h_{adv}^i$ denotes the worst-case affine transformation for node $i$ in the maximization problem (2); $\Pr_i$ denotes the probability measured by the underlying distribution of node $i$, and $f_{\boldsymbol{w}}(\mathbf{x})[j]$ denotes the output of the neural network's last softmax layer for label $j$. Note that for $\gamma = 0$, the above definition reduces to the average misclassification rate under the distribution shifts, which we simply denote by $\mathcal{L}^{\mathrm{adv}}(\boldsymbol{w})$. We also use $\hat{\mathcal{L}}_\gamma^{\mathrm{adv}}(\boldsymbol{w})$ to denote the above margin risk for the empirical distribution of samples, where we replace the underlying $\Pr_i$ with $\hat{\Pr}_i$ being the empirical probability evaluated for the $m$ samples of node $i$. The following theorem bounds the difference of the empirical and underlying margin-based error measures in (6) for a general deep neural network function.

**Theorem 3.** *Consider an L-layer neural network with $d$ neurons per layer. We assume the activation function of the neural network $\sigma$ satisfies $\sigma(0) = 0$ and $\max_t\{|\sigma'(t)|, |\sigma''(t)|\} \leq 1$. Suppose the same Lipschitzness and smoothness condition holds for loss $\ell$, and $\|\mathbf{X}\|_2 \leq B$. We assume the weights of the neural network are spectrally regularized such that for $M > 0$: $\frac{1}{M} \leq (\prod_{i=1}^d \|\boldsymbol{w}_i\|_\sigma)^{1/d} \leq M$ with $\|\cdot\|_\sigma$ denoting the spectral norm. Also, suppose that for $\eta > 0$, $\mathrm{Lip}(\nabla f_{\boldsymbol{w}}) := \sum_{i=1}^d \prod_{j=1}^i \|\boldsymbol{w}_i\|_\sigma \leq \lambda(1 - \eta)$ holds where $\mathrm{Lip}(\nabla f_{\boldsymbol{w}})$ upper-bounds the Lipschitz coefficient of the gradient $\nabla_{\mathbf{x}}\ell(f_{\boldsymbol{w}}(\mathbf{x}, y))$. Then, for every $\xi > 0$ with probability at least $1 - \xi$ the following holds for all feasible weights $\boldsymbol{w}$:*

$$\mathcal{L}^{\mathrm{adv}}(\boldsymbol{w}) - \hat{\mathcal{L}}_\gamma^{\mathrm{adv}}(\boldsymbol{w}) \leq \mathcal{O}\left(\sqrt{\frac{B^2 L^2 d \log(Ld)\lambda^2\big(\prod_{i=1}^L \|\boldsymbol{w}_i\|_\sigma \sum_{i=1}^L \frac{\|\boldsymbol{w}_i\|_F^2}{\|\boldsymbol{w}_i\|_\sigma^2}\big)^2 + L\log\frac{nmL\log(M)}{\eta\xi}}{m\gamma^2(\lambda - (1+B)\mathrm{Lip}(\nabla f_{\boldsymbol{w}}))^2}}\right).$$

This theorem gives a non-asymptotic bound on the generalization risk of `FLRA` for spectrally regularized neural nets with their smoothness constant bounded by $\lambda$. Thus, we can control the generalization performance by properly regularizing the Lipschitzness and smoothness degrees of the neural net.

### 4.3 Distributional robustness

To analyze `FLRA`'s robustness properties, we draw a connection between `FLRA` and distributionally robust optimization using optimal transport costs. Consider the optimal transport cost $W_c(P, Q)$ for quadratic cost $c(\mathbf{x}, \mathbf{x}') = \frac{1}{2}\|\mathbf{x} - \mathbf{x}'\|_2^2$ defined as $W_c(P, Q) := \min_{M \in \Pi(P,Q)} \mathbb{E}[c(\mathbf{X}, \mathbf{X}')]$, where $\Pi(P, Q)$ denotes the set of all joint distributions on $(\mathbf{X}, \mathbf{X}')$ with marginal distributions $P, Q$. In other words, $W_c(P, Q)$ measures the minimum expected cost for transporting samples between $P$ and $Q$. In order to define a distributionally robust federated learning problem against affine distribution shifts, we consider the following minimax problem:

$$\min_{\boldsymbol{w}} \frac{1}{n} \sum_{i=1}^n \max_{\Lambda^i, \delta^i} \left\{\mathbb{E}_{P^i}\big[\ell\big(f_{\boldsymbol{w}}(\Lambda^i\mathbf{X} + \delta^i), Y\big)\big] - W_c(P_{\mathbf{X}}^i, P_{\Lambda^i\mathbf{X}+\delta^i}^i)\right\}. \tag{7}$$

In this distributionally robust learning problem, we include a penalty term controlling the Wasserstein cost between the original distribution of node $i$ denoted by $P^i$ and its perturbed version under an affine distribution shift, i.e., $P_{\Lambda^i\mathbf{X}+\delta^i}^i$. Note that here we use the averaged Wasserstein cost $\frac{1}{n}\sum_{i=1}^n W_c(P_{\mathbf{X}}^i, P_{\Lambda^i\mathbf{X}+\delta^i}^i)$ to measure the distribution shift caused by the affine shifts $(\Lambda^i, \delta^i)_{i=1}^n$. The following theorem shows that this Wasserstein cost can be upper-bounded by a norm-squared function of $\Lambda$ and $\delta$ that appears in the `FLRA`'s minimax problem.

**Theorem 4.** *Consider the Wasserstein cost $W_c(P_{\mathbf{X}}, P_{\Lambda\mathbf{X}+\delta})$ between the distributions of $\mathbf{X}$ and its affine perturbation $\Lambda\mathbf{X} + \delta$. Assuming $\|\mathbb{E}[\mathbf{X}\mathbf{X}^T]\|_\sigma \leq \lambda$, we have*

$$W_c(P_{\mathbf{X}}, P_{\Lambda\mathbf{X}+\delta}) \leq \max\{\lambda, 1\}\big[\|\Lambda - I\|_F^2 + \|\delta\|_2^2\big]. \tag{8}$$

Substituting the Wasserstein cost in (7) with the upper-bound (8) results in the `FLRA`'s minimax (3). As a result, if $\frac{\lambda}{n}\sum_{i=1}^n[\|\Lambda^i - I\|_F^2 + \|\delta^i\|_2^2] \leq \varepsilon^2$ holds for the optimized $\Lambda^i, \delta^i$'s, we will also have the averaged Wasserstein cost bounded by $\frac{1}{n}\sum_{i=1}^n W_c(P_{\mathbf{X}}^i, P_{\Lambda^i\mathbf{X}+\delta^i}^i) \leq \varepsilon^2$. Theorem 4, therefore, shows the `FLRA`'s minimax approach optimizes a lower-bound on the distributionally robust (7).

## 5 Numerical Results

We implemented `FedRobust` in the Tensorflow platform [37] and numerically evaluated the algorithm's robustness performance against affine distribution shifts and adversarial perturbations. We considered the standard MNIST [38] and CIFAR-10 [39] datasets and used three standard neural network architectures in the literature: AlexNet [40], Inception-Net [41], and a mini-ResNet [42].

In the experiments, we simulated a federated learning scenario with $n = 10$ nodes where each node observes $m = 5000$ training samples. We manipulated the training samples at each node via an affine distribution shift randomly generated according to a Gaussian distribution. We also used 5000 test samples for which we did not apply any random affine shift and instead considered the following two scenarios: (1) affine distribution shifts by optimizing the inner maximization in (1) using projected gradient descent (PGD); (2) $\ell_2$-norm bounded adversarial PGD perturbations. We considered three

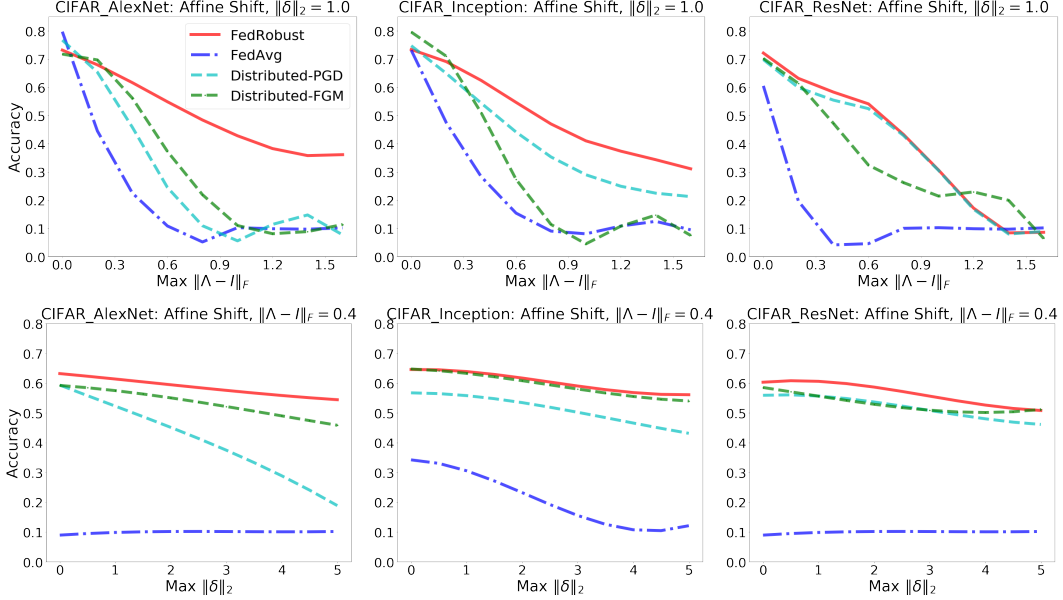

Figure 1: Test accuracy under affine distribution shifts over CIFAR-10. Top: constraining $\|\delta\|_2 \le 1$ and changing maximum allowed $\|\Lambda - I\|_F$. Bottom: constraining $\|\Lambda - I\|_F \le 0.4$ and changing maximum allowed $\|\delta\|_2$.

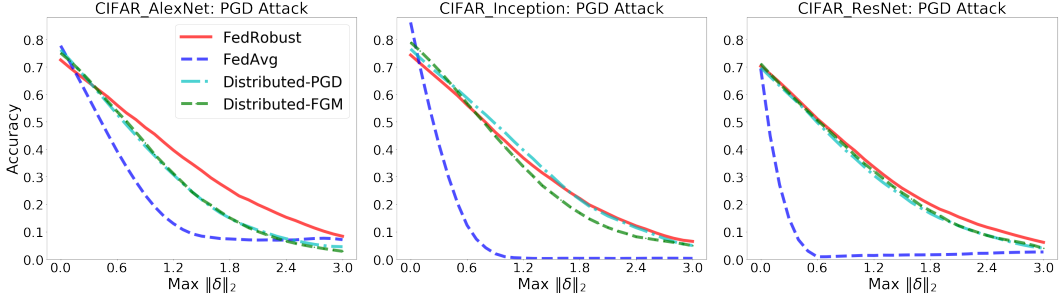

Figure 2: Test accuracy under PGD over CIFAR-10. $X$-axis shows the maximum allowed $\ell_2$-norm for PGD.

baselines: (1) `FedAvg` where the server node averages the updated parameters of the local nodes after every gradient step; (2) distributed FGM training where the nodes perform fast adversarial training [9] by optimizing a norm-bounded perturbation $\delta_j^i$ using one gradient step followed by projection onto an $\ell_2$-norm ball; (3) distributed PGD training where each node preforms PGD adversarial training [8] by applying 10 gradient steps where each step is followed by a projection onto an $\ell_2$-norm ball.

### 5.1 `FedRobust` vs. `FedAvg` and adversarial training: affine distribution Shifts

We tested the performance of the neural net classifiers trained by `FedRobust`, `FedAvg`, distributed FGM, and distributed PGD under different levels of affine distribution shifts. Figure 1 shows the accuracy performance over CIFAR-10 with AlexNet, Inception-Net, and ResNet architectures. As demonstrated, `FedRobust` outperforms the baseline methods in most of the experiments. The improvement over `FedAvg` can be as large as $54\%$. Moreover, `FedRobust` improved over distributed FGM and PGD adversarial training, which suggests adversarial perturbations may not be able to capture the complexity of affine distribution shifts. `FedRobust` also results in $4\times$ faster training compared to distributed PGD. These improvements motivate `FedRobust` as a robust and efficient federated learning method to protect against affine distribution shifts.

### 5.2 `FedRobust` vs. `FedAvg` and adversarial training: Adversarial perturbations

Figure 2 summarizes our numerical results of `FedRobust` and other baselines over CIFAR-10 where the plots show the test accuracy under different levels of $\ell_2$-norm perturbations. While we motivated `FedRobust` as a federated learning scheme protecting against affine distribution shifts, we

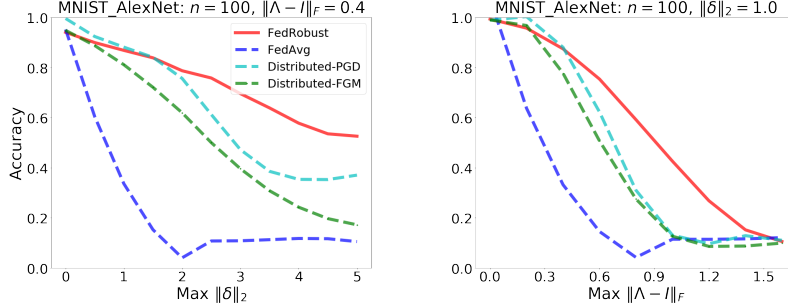

Figure 3: Test accuracy under affine perturbations for $n = 100$ nodes over MNIST data. $X$-axis shows the maximum allowed $\|\Lambda - I\|_F$ (left) and $\|\delta\|_2$ (right) for affine perturbations.

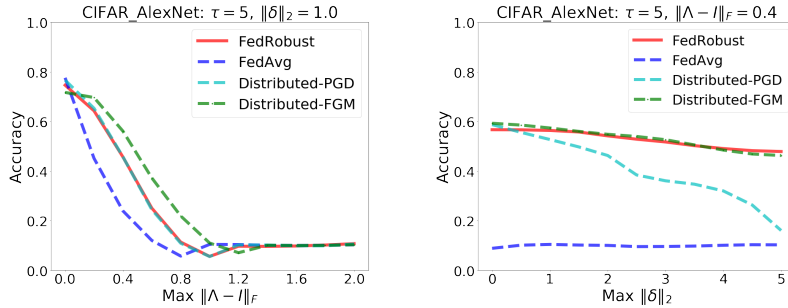

Figure 4: Test accuracy under affine perturbations for $\tau = 5$ minimization iteration count over CIFAR-10 data. $X$-axis shows the maximum allowed $\|\Lambda - I\|_F$ (left) and $\|\delta\|_2$ (right) for affine perturbations.

empirically observed its robust performance against adversarial perturbations as well. The achieved adversarial robustness in almost all cases matches the robustness offered by distributed FGM and PGD adversarial training. This observation can be explained by analyzing the generalization properties of these algorithms. We note that `FedRobust`'s improved robustness is obtained over the *test* samples. On the other hand, PGD consistently outperformed `FedRobust` on the *training* samples, achieving a near perfect training accuracy. However, `FedRobust` generalized better to the test samples and could overall outperform PGD on the test set. Also, the similar performance of FGM and PGD can be explained via the random Gaussian perturbations used for simulating the heterogeneity across clients and the results of [43] indicating FGM initialized at random perturbations performs as well as PGD. These numerical results indicate that affine distribution shifts can cover the distribution changes caused by norm-bounded adversarial perturbations. In summary, our numerical experiments demonstrate the efficiency and robustness of `FedRobust` against PGD adversarial attacks. We defer more details of our experiments and the numerical results on MNIST data to the Appendix.

Finally, we performed additional numerical experiments to analyze the effect of network size $n$ and minimization iteration count $\tau$ on the robustness performance. Figure 3 shows the results of our experiments for a larger network size of $n = 100$ AlexNet neural network classifiers, each trained using $m = 500$ MNIST training data points. As demonstrated in Figure 3's plots, `FedRobust` still outperforms the standard and adversarial training baselines over a wide range of affine perturbation parameters. To examine the effect of parameter $\tau$, i.e., minimization step count per training iteration, on our experimental results, we performed the CIFAR-10 experiment with the AlexNet architecture for $\tau = 5$ as demonstrated in Figure 4. We observed that after increasing $\tau$ to 5, the robustness offered by `FedRobust` slightly decreased and was comparable to the performance of our adversarial training baselines. While `FedRobust` still outperforms `FedAvg` by a clear margin, the numerical results indicate the role of simultaneous min-max optimization and proper selection of hyperparameters in the success of `FedRobust`.

We conclude this section by reiterating the practicality of the considered affine model. As demonstrated in our experiments, the affine model considered in this paper is particularly practical for image classification tasks in federated learning, where each camera's imperfections affect its pictures [7]. While this model provides significant robustness compared to additive-only perturbation models (i.e. $\Lambda = I$), it lays out potential new directions to study more complicated (non-affine) models such as neural network transformations.

## Broader Impact

As the amount of private data generated at users' devices surges, we are observing ever-growing interests in the industry to develop applications that leverage such personal and private data to boost the performance of their product, such as smart healthcare, online banking, self-driving cars, semantic learning, etc. To address such critical concern on users' privacy, governments in the U.S. and Europe have been passing regulations to ensure data protection and data traceability on such frameworks. Federated Learning is a novel learning paradigm that significantly improves data protection guarantees over the standard frameworks, in addition to various other upsides. In machine learning community in particular, developing federated learning and in general privacy preserving algorithms has gained compelling interest while user privacy is being acknowledged as an ethic in algorithm design.

This powerful paradigm is yet prone to several drawbacks and demands extensive theoretical and empirical studies. Our work particularly targets data heterogeneity and local resource consumption as two major concerns in federated learning methods. Our methodology results in a fast and robust federated learning framework and improves the performance of standard techniques. Image recognition is a particular use-case of our framework where we capture the variations of images due to camera imperfections or weather conditions impacting the images. Our robust method can improve the accuracy of a variety of image recognition tasks such as guiding autonomous robots, self-driving cars and accident avoidance systems, while protecting users' private data.

## Acknowledgment

The authors acknowledge supports from ONR grant N00014-20-1-2394, MIT-IBM Watson AI lab, NSF grants CNS 2003035, CCF 1909320, and UC Office of President Grant LFR-18-548175.

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
