[Supplementary Material · 6793_supplementary.pdf]

Supplementary material for paper:

**Robust Federated Learning: The Case of Affine Distribution Shifts**

## Appendix A   Additional Numerical Experiments

### A.1   Experimental Setup

In the experiments, we simulated a federated learning scenario with $n = 10$ nodes where each node observes $m = 5000$ training samples. We also divided the extra $10,000$ samples in each dataset to two validation and test sets containing $5000$ samples each. For CIFAR-10 samples, we applied the sandard normalization and scaled and linearly mapped the pixel intensity values to interval $[-1, 1]$. We applied batch normalization [44] in order to stabilize training and used the ADAM optimizer [45] with stepsize value $10^{-4}$ and default beta parameters $\beta_1 = 0.9$ and $\beta_2 = 0.99$ to optimize the neural net's parameters for $T = 100$ epochs (10000 iterations).

We did cross validation to choose $\lambda \in \{0.1, 0.5, 1, 5, 10, 50\}$ and chose the $\lambda$-value resulting in the closest additive penalty $\frac{1}{n} \sum_{i=1}^{n} [\|\Lambda^{i^*} - I\|_2^2 + \|\delta^{i^*}\|_2^2]$ to 10 percent of the average sample norm, i.e. $\frac{0.1}{m} \sum_{i=1}^{m} \|\mathbf{x}_i^{\mathrm{val}}\|_2^2$, over the $m = 5000$ validation samples. To perform GDA optimization, we applied two ascent steps per descent step with stepsize $\frac{1}{2\lambda}$. In order to simulate an affine distribution shift, we manipulated each $\tilde{\mathbf{x}}_j^i$ in the original training dataset via an affine transformation chosen randomly at each node:

$$\mathbf{x}_j^i = (I_d + \tilde{\Lambda}^i)\tilde{\mathbf{x}}_j^i + \tilde{\delta}^i. \tag{9}$$

Here, each $\tilde{\Lambda}^i$ is a random matrix with i.i.d. Gaussian entries according to $\mathcal{N}(0, \frac{\sigma^2}{d})$, and $\tilde{\delta}^i$ is a random Gaussian vector according to $\mathcal{N}(0, \sigma^2 I_d)$ where we set $\sigma = 0.01$. In test time, we did not apply any random affine transformation to test samples and instead considered the following three scenarios: (1) no perturbation, (2) adversarial affine distribution shift obtained by optimizing the inner maximization in (1) using projected gradient descent, 3) adversarial perturbations designed by the projected gradient descent algorithm. We used 100 projected gradient steps with stepsize $0.1$.

We considered three baselines in the experiments: (1) `FedAvg` where the server node averages the updated parameters of the local nodes after every gradient step, (2) Distributed FGM training where the nodes perform fast adversarial training [9] by optimizing an $\ell_2$-norm bounded perturbation $\delta_j^i$ using one gradient step followed by projection onto the ball $\{\delta_j^i : \|\delta_j^i\|_2 \leq \epsilon_{\mathrm{fgm}}\}$, and (3) Distributed PGD training where each node preforms PGD adversarial training [8] similar to distributed FGM but uses 10 projected gradient steps, each followed by projection onto $\{\delta_j^i : \|\delta_j^i\|_2 \leq \epsilon_{\mathrm{pgd}}\}$. We used the value $\epsilon_{\mathrm{fgm}} = \epsilon_{\mathrm{pgd}} = 0.05\mathbb{E}[\|\mathbf{x}_i\|_2]$ in the experiments. We observed training instability after achieving perfect training accuracy for the baseline `FedAvg` algorithm, and hence performed early stopping to avoid the instability in the `FedAvg` experiments. We did not encounter the instability issue in `FedRobust` experiments.

### A.2   Numerical Results for MNIST data

We repeated the CIFAR experiments in Figures 1 and 2 for the MNIST dataset. Figure 5 shows the numerical results under affine distribution shifts. The figure's top row includes the plots for fixed maximum delta norm $\|\delta\|_2 \leq 1$ and different levels of maximum allowed $\|\Lambda - I\|_F$, while in the bottom row we fix the maximum allowed linear shift $\|\Lambda - I\|_F \leq 0.6$ and evaluate the test accuracy under different levels of $\|\delta\|_2$. As shown in the plots, `FedRobust` results in the best performance in most of the evaluations, which indicates the superior performance of `FedRobust` against affine distribution shifts. Figure 6 shows the test accuracy of the trained networks under different levels of adversarial PGD perturbations. The figure's experiments again shows that `FedRobust` can effectively shield against PGD adversarial attacks and achieve a comparable performance to PGD and FGM adversarial training.

Figure 5: Trained networks' test accuracy under affine distribution shifts in the MNIST experiments. Top row: constraining $\|\delta\|_2 \leq 1$ and changing maximum allowed $\|\Lambda - I\|_F$, bottom row: constraining $\|\Lambda - I\|_F \leq 0.6$ and changing maximum allowed $\|\delta\|_2$.

Figure 6: Trained networks' test accuracy under PGD perturbations in the MNIST experiments. $X$-axis shows the maximum allowed $\ell_2$-norm for PGD perturbations.

# Appendix B   Preliminaries and Useful Lemmas

In this section, we provide preliminary and useful results in order to prove Theorems 1 and 2. For notational convenience, we use the following short-hand notations:

| Notation | Description |
|---|---|
| $\boldsymbol{\psi}_t^i = \left(\Lambda^i,\, \delta_t^i\right)$ | maximization variables of node $i$ iteration $t$ |
| $\Psi_t = \left(\boldsymbol{\psi}_t^1;\, \cdots;\, \boldsymbol{\psi}_t^n\right)$ | concatenation of all nodes' maximization models at iteration $t$ |
| $\overline{\boldsymbol{w}}_t = \dfrac{1}{n}\sum_{i\in[n]} \boldsymbol{w}_t^i$ | average model at iteration $t$ |
| $a_t = \mathbb{E}\left[\Phi(\overline{\boldsymbol{w}}_t)\right] - \Phi^*$ | optimality gap measure between $\Phi(\overline{\boldsymbol{w}}_t)$ and $\min_{\boldsymbol{w}} \Phi(\boldsymbol{w})$ |
| $b_t = \mathbb{E}\left[\Phi(\overline{\boldsymbol{w}}_t) - f(\overline{\boldsymbol{w}}_t, \Psi_t)\right]$ | optimality gap measure between $f(\overline{\boldsymbol{w}}_t, \Psi_t)$ and $\max_{\Psi} f(\overline{\boldsymbol{w}}_t, \Psi)$ |
| $e_t = \dfrac{1}{n}\sum_{i\in[n]} \mathbb{E}\left\|\boldsymbol{w}_t^i - \overline{\boldsymbol{w}}_t\right\|^2$ | average deviation of the local models from the average model at iteration $t$ |
| $g_t = \mathbb{E}\left\|\dfrac{1}{n}\sum_{i\in[n]} \nabla_{\boldsymbol{w}} f^i(\boldsymbol{w}_t^i, \boldsymbol{\psi}_t^i)\right\|^2$ | norm squared of local gradients w.r.t $\boldsymbol{w}$ at iteration $t$ |
| $h_t = \mathbb{E}\left\|\nabla\Phi(\overline{\boldsymbol{w}}_t) - \dfrac{1}{n}\sum_{i\in[n]} \nabla_{\boldsymbol{w}} f^i(\boldsymbol{w}_t^i, \boldsymbol{\psi}_t^i)\right\|^2$ | norm squared of deviation in gradients w.r.t $\boldsymbol{w}$ of $\max_{\Psi} f(\overline{\boldsymbol{w}}_t, \Psi)$ and local functions $f^i(\boldsymbol{w}_t^i, \boldsymbol{\psi}_t^i)$ |

Table 1: Table of notations.

Now, we present a set of useful lemmas and observations which we will invoke to prove the convergence results for both PL-PL and nonconvex-PL loss cases. The following lemma establishes the Lipschitz gradient parameter for the global function given those of the local objectives.

**Lemma 1.** *If the local functions $f^i$s have Lipschits gradients with parameters stated in Assumption 3, then the global function $f$ has also Lipschitz gradients as follows: for any $\boldsymbol{w}, \boldsymbol{w}', \Psi, \Psi'$ it holds that*

$$\left\|\nabla_{\boldsymbol{w}} f(\boldsymbol{w}, \Psi) - \nabla_{\boldsymbol{w}} f(\boldsymbol{w}', \Psi)\right\| \le L_1\left\|\boldsymbol{w} - \boldsymbol{w}'\right\|, \left\|\nabla_{\boldsymbol{w}} f(\boldsymbol{w}, \Psi) - \nabla_{\boldsymbol{w}} f(\boldsymbol{w}, \Psi')\right\| \le \frac{L_{12}}{\sqrt{n}}\left\|\Psi - \Psi'\right\|_F,$$

$$\left\|\nabla_{\Psi} f(\boldsymbol{w}, \Psi) - \nabla_{\Psi} f(\boldsymbol{w}', \Psi)\right\|_F \le \frac{L_{21}}{\sqrt{n}}\left\|\boldsymbol{w} - \boldsymbol{w}'\right\|, \left\|\nabla_{\Psi} f(\boldsymbol{w}, \Psi) - \nabla_{\Psi} f(\boldsymbol{w}, \Psi')\right\|_F \le \frac{L_2}{n}\left\|\Psi - \Psi'\right\|_F.$$

(10)

*Proof.* We defer the proof to Section E.1. $\square$

Recall the definition of the function $\Phi(\cdot)$, that is,

$$\Phi(\boldsymbol{w}) := \max_{\Psi} f(\boldsymbol{w}, \Psi) = \max_{\boldsymbol{\psi}^1, \cdots, \boldsymbol{\psi}^n} \frac{1}{n}\sum_{i\in[n]} f^i(\boldsymbol{w}, \boldsymbol{\psi}^i) = \max_{(\Lambda^1, \delta^1), \cdots, (\Lambda^n, \delta^n)} \frac{1}{n}\sum_{i\in[n]} f^i(\boldsymbol{w}, \Lambda^i, \delta^i). \quad (11)$$

Next lemma shows that $\Phi$ has Lipschitz gradients and characterizes its parameter.

**Lemma 2** ([32]). *If Assumptions 3 and 4 (ii) hold, that is, the local objectives have Lipschitz gradients and $-f(\boldsymbol{w}, \cdot)$ is $\mu_2$-PL, then we have*

$$\nabla\Phi(\boldsymbol{w}) = \nabla_{\boldsymbol{w}} f(\boldsymbol{w}, \Psi^*(\boldsymbol{w})), \quad (12)$$

*where $\Psi^*(\boldsymbol{w}) \in \arg\max_{\Psi} f(\boldsymbol{w}, \Psi)$ for any $\boldsymbol{w}$. Moreover, $\Phi$ has Lipschitz gradients with parameter $L_\Phi = L_1 + \frac{L_{12}L_{21}}{2n\mu_2}$.*

*Proof.* We defer the proof to Section E.2. □

Next lemma shows the contraction of the sequence $\{\mathbb{E}[\Phi(\overline{\boldsymbol{w}}_t)]\}_{t\geq 0}$ when running the update rule of FedRobust method in Algorithm 1. Please refer to Table 1 to recall the definition of $h_t$ and $g_t$.

**Lemma 3.** *If Assumptions 2 and 3 hold, then the iterates of* FedRobust *satisfy the following contraction inequality for any iteration $t \geq 0$*

$$\mathbb{E}[\Phi(\overline{\boldsymbol{w}}_{t+1})] - \mathbb{E}[\Phi(\overline{\boldsymbol{w}}_t)] \leq -\frac{\eta_1}{2}\mathbb{E}\|\nabla\Phi(\overline{\boldsymbol{w}}_t)\|^2 + \frac{\eta_1}{2}h_t - \frac{\eta_1}{2}(1-\eta_1 L_\Phi)g_t + \eta_1^2\frac{L_\Phi}{2}\frac{\sigma_{\boldsymbol{w}}^2}{n}. \quad (13)$$

*Proof.* We defer the proof to Section E.3. □

Next lemma further bounds $h_t$ w.r.t. the two sequences $b_t$ and $e_t$.

**Lemma 4.** *If Assumptions 3 and 4 (ii) hold, that is, the local objectives have Lipschitz gradients and $-f(\boldsymbol{w}, \cdot)$ is $\mu_2$-PL, then we have*

$$h_t \leq \frac{4L_{12}^2}{\mu_2 n}b_t + 2L_1^2 e_t. \quad (14)$$

*Proof.* We defer the proof to Section E.4. □

Next lemma establishes a contraction bound on the sequence $b_t$.

**Lemma 5.** *If Assumptions 2, 3 and 4 (ii) hold, then the sequence of $\{b_t\}_{t\geq 0}$ generated by the* FedRobust *iterations with $\eta_2 \leq 1/L_2$ satisfies the following contraction bound:*

$$b_{t+1} \leq (1-\mu_2\eta_2 n)\left(1+\eta_1\frac{4L_{12}^2}{\mu_2 n}\right)b_t + \frac{\eta_1}{2}\mathbb{E}\|\nabla\Phi(\overline{\boldsymbol{w}}_t)\|^2 + \frac{\eta_1^2}{2}\left(L_1 + L_\Phi + 2\eta_2 L_{21}^2\right)g_t$$

$$+ \left(\eta_1 L_1^2 + \eta_2 L_{21}^2\right)e_t + \frac{\eta_1^2}{2}\left(L_1 + L_\Phi + 2\eta_2 L_{21}^2\right)\frac{\sigma_{\boldsymbol{w}}^2}{n} + \frac{\eta_2^2}{2}L_2\sigma_\psi^2, \quad (15)$$

*where $L_\Phi$ is the Lipschitz gradient parameter of the function $\Phi(\cdot)$ characterized in Lemma 2.*

*Proof.* We defer the proof to Section E.5. □

Next lemma bounds $e_t$, that is the average deviation of local parameter models from their average.

**Lemma 6.** *If Assumptions 1, 2 and 3 hold and the step-size $\eta_1$ satisfies $32\eta_1^2(\tau-1)^2 L_1^2 \leq 1$, then the sequence $e_t = \frac{1}{n}\sum_{i\in[n]}\mathbb{E}\|\boldsymbol{w}_t^i - \overline{\boldsymbol{w}}_t\|^2$ is bounded as follows*

$$e_t \leq 16\eta_1^2(\tau-1)^2\rho^2 + 4\eta_1^2(\tau-1)(n+1)\frac{\sigma_{\boldsymbol{w}}^2}{n} + 20\eta_1^2(\tau-1)\sum_{l=t_c+1}^{t-1}g_l, \quad (16)$$

*where $t_c$ denotes the index of the most recent server-worker communication, i.e. $t_c = \lfloor\frac{t}{\tau}\rfloor\tau$ and we also denote $\rho^2 := 3\rho_f^2 + 6L_{12}^2(\epsilon_1^2 + \epsilon_2^2)$.*

*Proof.* We defer the proof to Section E.6. □

Next generic lemma is adopted form [16].

**Lemma 7.** *Assume that two non-negative sequences $\{P_t\}_{t\geq 0}$ and $\{g_t\}_{t\geq 0}$ satisfy the following inequality for each iteration $t \geq 0$ and some constants $0 < \Upsilon < 1$, $L \geq 0$, $B \geq 0$, and $\Gamma \geq 0$:*

$$P_{t+1} \leq \Upsilon P_t - \frac{\eta_1}{2}(1-\eta_1 L)g_t + \eta_1^2 B\sum_{l=t_c+1}^{t-1}g_l + \Gamma, \quad (17)$$

*where $t_c = \lfloor\frac{t}{\tau}\rfloor\tau$. Then, for each $t \geq 0$ we have*

$$P_t \leq \Upsilon^t P_0 + \frac{\Gamma}{1-\Upsilon}, \quad (18)$$

*if $\eta_1$ satisfies the following condition*

$$\eta_1 \left( L + \frac{2B}{\Upsilon^{\tau-1}(1-\Upsilon)} \right) \leq 1. \tag{19}$$

*Proof.* We defer the proof to Section E.7. □

Next lemma bounds the overall optimality gap $b_t$ averaged over $T$ iterations.

**Lemma 8.** *If Assumptions 2, 3 and 4 (ii) hold and the step-sizes satisfy the conditions $\eta_2 \leq 1/L_2$ and $\frac{\eta_2}{\eta_1} \geq \frac{8L_{12}^2}{\mu_2^2 n^2}$, then the average of the sequence $\{b_t\}_{t=0}^{T-1}$ generated from the* FedRobust *can be bounded as follows:*

$$\frac{1}{T} \sum_{t=0}^{T-1} b_t \leq \frac{4L_2^2}{\mu_2^2 n^2} \frac{\epsilon_1^2 + \epsilon_2^2}{\eta_2 T} + \frac{\eta_1}{\eta_2} \frac{1}{\mu_2 n} \frac{1}{T} \sum_{t=0}^{T-1} \mathbb{E} \left\| \nabla \Phi(\overline{\boldsymbol{w}}_t) \right\|^2$$

$$+ \frac{\eta_1^2}{\eta_2} \frac{1}{\mu_2 n} \left( L_1 + L_\Phi + 2\eta_2 L_{21}^2 \right) \frac{1}{T} \sum_{t=0}^{T-1} g_t + \frac{1}{\eta_2} \frac{2}{\mu_2 n} \left( \eta_1 L_1^2 + \eta_2 L_{21}^2 \right) \frac{1}{T} \sum_{t=0}^{T-1} e_t$$

$$+ \frac{\eta_1^2}{\eta_2} \frac{1}{\mu_2 n} \left( L_1 + L_\Phi + 2\eta_2 L_{21}^2 \right) \frac{\sigma_{\boldsymbol{w}}^2}{n} + \eta_2 \frac{L_2}{\mu_2 n} \sigma_\psi^2, \tag{20}$$

*where $L_\Phi$ is the Lipschitz gradient parameter of the function $\Phi(\cdot)$ characterized in Lemma 2 and $\epsilon_1, \epsilon_2$ represent the radius of the affine perturbation balls, i.e. $\|\Lambda^i - I\| \leq \epsilon_1$ and $\|\delta^i\| \leq \epsilon_2$ for each node $i \in [n]$.*

*Proof.* We defer the proof to Section E.8. □

Next lemma bounds the averaged local model deviations $e_t$ over $T$ iterations.

**Lemma 9.** *If Assumptions 1, 2 and 3 hold and the step-size $\eta_1$ satisfies $32\eta_1^2(\tau-1)^2 L_1^2 \leq 1$, then the average of the sequence $e_t$ over $t = 0, \cdots, T-1$ is bounded as follows*

$$\frac{1}{T} \sum_{t=0}^{T-1} e_t \leq 20\eta_1^2(\tau-1)^2 \frac{1}{T} \sum_{t=0}^{T-1} g_t + 16\eta_1^2(\tau-1)^2 \rho^2 + 8\eta_1^2(\tau-1)(n+1) \frac{\sigma_{\boldsymbol{w}}^2}{n}. \tag{21}$$

*Proof.* We defer the proof to Section E.9. □

## Appendix C    Proof of Theorem 1

Having established the key lemmas, now we proceed to prove Theorem 1 for any $\beta \leq 1/2$. To show the convergence of the sequence $P_t = a_t + \beta b_t$, we firstly need to establish a contraction inequality on $P_{t+1}$ with respect to $P_t$. We begin by the following bound on the sequence $a_t = \mathbb{E}[\Phi(\overline{\boldsymbol{w}}_t)] - \Phi^*$ which is directly implied from Lemma 3:

$$a_{t+1} \leq a_t - \frac{\eta_1}{2} \mathbb{E} \left\| \nabla \Phi(\overline{\boldsymbol{w}}_t) \right\|^2 + \frac{\eta_1}{2} h_t - \frac{\eta_1}{2} \left( 1 - \eta_1 L_\Phi \right) g_t + \eta_1^2 \frac{L_\Phi}{2} \frac{\sigma_{\boldsymbol{w}}^2}{n}. \tag{22}$$

Using Lemma 4 that shows $h_t \leq 4L_{12}^2 b_t / (\mu_2 n) + 2L_1^2 e_t$, the bound in (22) yields that

$$a_{t+1} \leq a_t - \frac{\eta_1}{2} \mathbb{E} \left\| \nabla \Phi(\overline{\boldsymbol{w}}_t) \right\|^2 + \eta_1 \frac{2L_{12}^2}{\mu_2 n} b_t + \eta_1 L_1^2 e_t - \frac{\eta_1}{2} \left( 1 - \eta_1 L_\Phi \right) g_t + \eta_1^2 \frac{L_\Phi}{2} \frac{\sigma_{\boldsymbol{w}}^2}{n}. \tag{23}$$

Next, we employ the result of Lemma 5 which establishes a contraction bound on the $b_t$ sequence. Putting together with (23) implies that

$$P_{t+1} = a_{t+1} + \beta b_{t+1}$$

$$\leq a_t - \frac{\eta_1}{2}\left(1 - \beta\right)\mathbb{E}\big\|\nabla\Phi(\overline{\boldsymbol{w}}_t)\big\|^2$$

$$+ \beta\left(\eta_1\frac{2L_{12}^2}{\beta\mu_2 n} + (1 - \mu_2\eta_2 n)\left(1 + \eta_1\frac{4L_{12}^2}{\mu_2 n}\right)\right)b_t$$

$$- \left(\frac{\eta_1}{2}\left(1 - \eta_1 L_\Phi\right) - \eta_1^2\frac{\beta}{2}\left(L_1 + L_\Phi + 2\eta_2 L_{21}^2\right)\right)g_t$$

$$+ \left(\eta_1 L_1^2 + \beta\left(\eta_1 L_1^2 + \eta_2 L_{21}^2\right)\right)e_t$$

$$+ \frac{\eta_1^2}{2}\left(L_\Phi + \beta\left(L_1 + L_\Phi + 2\eta_2 L_{21}^2\right)\right)\frac{\sigma_{\boldsymbol{w}}^2}{n} + \eta_2^2 L_2\frac{\beta}{2}\sigma_\psi^2. \quad (24)$$

We begin simplifying the above bound by first considering the first two terms in RHS of (24). We can show that the function $\Phi(\cdot)$ is $\mu_1$-PL [31], which implies that

$$\mathbb{E}\big\|\nabla\Phi(\overline{\boldsymbol{w}}_t)\big\|^2 \geq 2\mu_1\mathbb{E}\big[\Phi(\overline{\boldsymbol{w}}_t)\big] - \Phi^* = 2\mu_1 a_t. \quad (25)$$

Therefore, for any $\beta \leq 1/2$ we have

$$a_t - \frac{\eta_1}{2}\left(1 - \beta\right)\mathbb{E}\big\|\nabla\Phi(\overline{\boldsymbol{w}}_t)\big\|^2 \leq \left(1 - \frac{1}{2}\mu_1\eta_1\right)a_t, \quad (26)$$

which implies the coefficient of $a_t$ in (24) is bounded by $1 - \frac{1}{2}\mu_1\eta_1$. Next, the coefficient of $\beta b_t$ in (24) can be bounded as follows:

$$\eta_1\frac{2L_{12}^2}{\beta\mu_2 n} + (1 - \mu_2\eta_2 n)\left(1 + \eta_1\frac{4L_{12}^2}{\mu_2 n}\right) = 1 - \eta_1\frac{L_1 L_2}{\mu_2 n}\left(\frac{\mu_2^2\eta_2 n}{\eta_1 L_1 L_2} - \frac{2L_{21}^2}{\beta L_1 L_2} - 4(1 - \mu_2\eta_2 n)\frac{L_{21}^2}{L_1 L_2}\right)$$

$$\overset{(a)}{\leq} 1 - \eta_1\frac{L_1 L_2}{\mu_2 n}$$

$$\overset{(b)}{\leq} 1 - \frac{1}{2}\mu_1\eta_1, \quad (27)$$

where $(a)$ holds for our choice of $\beta$ and assuming $\frac{\mu_2^2\eta_2 n}{\eta_1 L_1 L_2} \geq 1 + \left(4 + \frac{2}{\beta}\right)\frac{L_{12}^2}{L_1 L_2}$ and $(b)$ is implies from the fact that

$$\frac{\eta_1\frac{L_1 L_2}{\mu_2 n}}{\frac{1}{2}\mu_1\eta_1} = 2\left(\frac{L_1}{\mu_1}\right)\left(\frac{L_2}{\mu_2 n}\right) \geq 1. \quad (28)$$

Now that we have bounded the coefficients of $a_t$ and $\beta b_t$ in (24), rearranging the terms and using the assumption $\eta_2 \leq 1/L_2$ simplifies the contraction on $P_t$ as follows

$$P_{t+1} \leq \left(1 - \frac{1}{2}\mu_1\eta_1\right)P_t - \frac{\eta_1}{2}\left(1 - \eta_1\hat{L}_\beta\right)g_t + \tilde{L}_\beta e_t + \eta_1^2\frac{\hat{L}_\beta}{2}\frac{\sigma_{\boldsymbol{w}}^2}{n} + \eta_2^2\frac{L_2}{2}\beta\sigma_\psi^2, \quad (29)$$

where we picked the following notations for convenient of the exposition

$$\tilde{L}_\beta = (1 + \beta)\eta_1 L_1^2 + \beta\eta_2 L_{21}^2, \quad \hat{L}_\beta = (1 + \beta)L_\Phi + \beta L_1 + 2\beta\frac{L_{21}^2}{L_2}. \quad (30)$$

Next, we use Lemma 6 which for $32\eta_1^2(\tau - 1)^2 L_1^2 \leq 1$ provides an upper bound on $e_t$ with respect to $g_t$. We can write

$$P_{t+1} \leq \left(1 - \frac{1}{2}\mu_1\eta_1\right)P_t - \frac{\eta_1}{2}\left(1 - \eta_1\hat{L}_\beta\right)g_t + 20\eta_1^2\tilde{L}_\beta(\tau - 1)\sum_{l=t_c+1}^{t-1} g_l$$

$$+ 16\eta_1^2\tilde{L}_\beta(\tau - 1)^2\rho^2 + 4\eta_1^2\tilde{L}_\beta(\tau - 1)(n + 1)\frac{\sigma_{\boldsymbol{w}}^2}{n} + \eta_1^2\frac{\hat{L}_\beta}{2}\frac{\sigma_{\boldsymbol{w}}^2}{n} + \eta_2^2\frac{L_2}{2}\beta\sigma_\psi^2. \quad (31)$$

We have shown in Lemma 7 that how a such contraction sequence converges. In particular, let us pick the following notations and apply the result of Lemma 7 to contraction in (31)

$$L = \hat{L}_\beta,$$

$$\Upsilon = 1 - \frac{1}{2}\mu_1\eta_1,$$

$$B = 20\tilde{L}_\beta(\tau - 1),$$

$$\Gamma = 16\eta_1^2\tilde{L}_\beta(\tau - 1)^2\rho^2 + 4\eta_1^2\tilde{L}_\beta(\tau - 1)(n + 1)\frac{\sigma_{\boldsymbol{w}}^2}{n} + \eta_1^2\frac{\hat{L}_\beta}{2}\frac{\sigma_{\boldsymbol{w}}^2}{n} + \eta_2^2\frac{L_2}{2}\beta\sigma_\psi^2. \tag{32}$$

It implies that if the step-sizes satisfy the following condition

$$\eta_1\left(\hat{L}_\beta + \frac{80\tilde{L}_\beta(\tau - 1)}{\eta_1\mu_1\left(1 - \frac{1}{2}\mu_1\eta_1\right)^{\tau-1}}\right) \le 1, \tag{33}$$

then we have

$$P_t \le \left(1 - \frac{1}{2}\mu_1\eta_1\right)^t P_0 + 32\eta_1\frac{\tilde{L}_\beta}{\mu_1}(\tau - 1)^2\rho^2 + 8\eta_1\frac{\tilde{L}_\beta}{\mu_1}(\tau - 1)(n + 1)\frac{\sigma_{\boldsymbol{w}}^2}{n} + \eta_1\frac{\hat{L}_\beta}{\mu_1}\frac{\sigma_{\boldsymbol{w}}^2}{n} + \frac{\eta_2^2}{\eta_1}\frac{L_2}{\mu_1}\beta\sigma_\psi^2, \tag{34}$$

which concludes the proof of Theorem 1. Note to hold this result, in addition to condition (33), we have assumed the following constraints on the step-sizes as well

$$\eta_2 L_2 \le 1, \quad 32\eta_1^2(\tau - 1)^2 L_1^2 \le 1, \quad \frac{\mu_2^2\eta_2 n}{\eta_1 L_1 L_2} \ge 1 + \left(4 + \frac{2}{\beta}\right)\frac{L_{12}^2}{L_1 L_2}. \tag{35}$$

## Appendix D  Proof of Theorem 2

We begin the proof by combining the results of Lemmas 3 and 4 which yields that for every iteration $t = 0, \cdots, T - 1$ we have

$$\mathbb{E}[\Phi(\overline{\boldsymbol{w}}_{t+1})] - \mathbb{E}[\Phi(\overline{\boldsymbol{w}}_t)] \le -\frac{\eta_1}{2}\mathbb{E}\|\nabla\Phi(\overline{\boldsymbol{w}}_t)\|^2 - \frac{\eta_1}{2}(1 - \eta_1 L_\Phi)g_t + \eta_1\frac{2L_{12}^2}{\mu_2 n}b_t + \eta_1 L_1^2 e_t + \eta_1^2\frac{L_\Phi}{2}\frac{\sigma_{\boldsymbol{w}}^2}{n}. \tag{36}$$

Summing up all the $T$ inequalities in (36) for $t = 0, \cdots, T - 1$ and dividing by $T$ yields the following

$$\frac{1}{T}\left(\mathbb{E}[\Phi(\overline{\boldsymbol{w}}_T)] - \Phi(\overline{\boldsymbol{w}}_0)\right) \le -\frac{\eta_1}{2}\frac{1}{T}\sum_{t=0}^{T-1}\mathbb{E}\|\nabla\Phi(\overline{\boldsymbol{w}}_t)\|^2$$

$$-\frac{\eta_1}{2}(1 - \eta_1 L_\Phi)\frac{1}{T}\sum_{t=0}^{T-1}g_t$$

$$+\eta_1\frac{2L_{12}^2}{\mu_2 n}\frac{1}{T}\sum_{t=0}^{T-1}b_t$$

$$+\eta_1 L_1^2\frac{1}{T}\sum_{t=0}^{T-1}e_t$$

$$+\eta_1^2\frac{L_\Phi}{2}\frac{\sigma_{\boldsymbol{w}}^2}{n}. \tag{37}$$

Next we use Lemmas 8 and then Lemma 9 to replace the terms $\frac{1}{T}\sum_{t=0}^{T-1} b_t$ and $\frac{1}{T}\sum_{t=0}^{T-1} e_t$ and rewrite the above bound in terms of $\frac{1}{T}\sum_{t=0}^{T-1} g_t$. It yields that

$$
\begin{aligned}
\frac{1}{T}\left(\mathbb{E}[\Phi(\overline{\boldsymbol{w}}_T)]-\Phi(\overline{\boldsymbol{w}}_0)\right) \leq & -\frac{\eta_1}{2}\left(1-\eta_1\frac{4L_{12}^2 L_2}{\mu_2^2 n^2}\right)\frac{1}{T}\sum_{t=0}^{T-1}\mathbb{E}\left\|\nabla\Phi(\overline{\boldsymbol{w}}_t)\right\|^2 \\
& -\frac{\eta_1}{2}\left(1-\eta_1\left(\hat{L}+40\tilde{L}(\tau-1)^2\right)\right)\frac{1}{T}\sum_{t=0}^{T-1} g_t \\
& +\frac{\eta_1}{\eta_2}\frac{8L_{12}^2 L_2^2}{\mu_2^3 n^3}\frac{\epsilon_1^2+\epsilon_2^2}{T}+16\eta_1^2\tilde{L}(\tau-1)^2\rho^2+\frac{\eta_1^2}{2}\hat{L}\frac{\sigma_{\boldsymbol{w}}^2}{n}+\eta_1\eta_2\frac{4L_{12}^2}{\mu_2^2 n^2}\hat{L}\sigma_\psi^2,
\end{aligned}
\tag{38}
$$

where we adopt the following short-hand notations

$$
\tilde{L}=\frac{3}{2}\eta_1 L_1^2+\frac{1}{2}\eta_2 L_{21}^2,\quad \hat{L}=\frac{3}{2}L_\Phi+\frac{1}{2}L_1+\frac{L_{21}^2}{L_2}.
\tag{39}
$$

Finally, we use the assumption $\eta_1(\hat{L}+40\tilde{L}(\tau-1)^2)\leq 1$ to remove the term $\frac{1}{T}\sum_{t=0}^{T-1} g_t$ and apply $\frac{\eta_1}{\eta_2}\leq\frac{\mu_2^2 n^2}{8L_{12}^2}$ to simply the bound and conclude the proof:

$$
\begin{aligned}
\frac{1}{T}\sum_{t=0}^{T-1}\mathbb{E}\left\|\nabla\Phi(\overline{\boldsymbol{w}}_t)\right\|^2 \leq & \frac{4\Delta_\Phi}{\eta_1 T}+\frac{4L_2^2}{\mu_2^2 n^2}\frac{\epsilon_1^2+\epsilon_2^2}{\eta_1 T}+64\eta_1\tilde{L}(\tau-1)^2\rho^2 \\
& +16\eta_1\tilde{L}(\tau-1)(n+1)\frac{\sigma_{\boldsymbol{w}}^2}{n}+2\eta_1\hat{L}\frac{\sigma_{\boldsymbol{w}}^2}{n}+\frac{\eta_2^2}{\eta_1}L_2\sigma_\psi^2.
\end{aligned}
\tag{40}
$$

## Appendix E    Proof of Useful Lemmas

### E.1    Proof of Lemma 1

Proof of all four cases in the claim is simple. We derive the proof for the fourth one as an instance. Recall definition of the global function $f$, that is

$$
f(\boldsymbol{w},\Psi)=\frac{1}{n}\sum_{i\in[n]} f^i(\boldsymbol{w},\boldsymbol{\psi}^i).
\tag{41}
$$

Therefore, the gradient of $f$ with respect to $\Psi$ is

$$
\nabla_\Psi f(\boldsymbol{w},\Psi)=\begin{pmatrix}\frac{\partial}{\partial\boldsymbol{\psi}^1} f(\boldsymbol{w},\Psi)\\ \vdots\\ \frac{\partial}{\partial\boldsymbol{\psi}^n} f(\boldsymbol{w},\Psi)\end{pmatrix}=\frac{1}{n}\begin{pmatrix}\nabla_\psi f^1(\boldsymbol{w},\boldsymbol{\psi}^1)\\ \vdots\\ \nabla_\psi f^n(\boldsymbol{w},\boldsymbol{\psi}^n)\end{pmatrix}.
\tag{42}
$$

We can then write for any $\boldsymbol{w}$, $\Psi=(\boldsymbol{\psi}^1;\cdots;\boldsymbol{\psi}^n)$, $\Psi'=(\boldsymbol{\psi}'^1;\cdots;\boldsymbol{\psi}'^n)$ and using Assumption 3 that

$$
\begin{aligned}
\left\|\nabla_\Psi f(\boldsymbol{w},\Psi)-\nabla_\Psi f(\boldsymbol{w},\Psi')\right\|_F^2 &=\frac{1}{n^2}\sum_{i\in[n]}\left\|\nabla_\psi f^i(\boldsymbol{w},\boldsymbol{\psi}^i)-\nabla_\psi f^i(\boldsymbol{w},\boldsymbol{\psi}'^i)\right\|_F^2 \\
&\leq\frac{L_2^2}{n^2}\sum_{i\in[n]}\left\|\boldsymbol{\psi}^i-\boldsymbol{\psi}'^i\right\|_F^2 \\
&=\frac{L_2^2}{n^2}\left\|\Psi-\Psi'\right\|_F^2.
\end{aligned}
\tag{43}
$$

### E.2    Proof of Lemma 2

The detailed proof can be found in [32], Lemma A.5. Note that in our case, according to Lemma 1 the function $f$ has Lipschitz gradients with constants $L_1, L_{12}/\sqrt{n}, L_{21}/\sqrt{n}, L_2/n$; implying the Lipschitz gradient parameter of the function $\Phi$ to be

$$
L_\Phi=L_1+\frac{(L_{12}/\sqrt{n})(L_{21}/\sqrt{n})}{2\mu_2}=L_1+\frac{L_{12}L_{21}}{2n\mu_2}.
\tag{44}
$$

### E.3 Proof of Lemma 3

We invoke Lemma 2 which shows that the gradient of the function $\Phi(\cdot)$ is $L_\Phi$-Lipschitz. We can write

$$
\begin{aligned}
\Phi(\overline{\boldsymbol{w}}_{t+1}) - \Phi(\overline{\boldsymbol{w}}_t) &\le \left\langle \nabla\Phi(\overline{\boldsymbol{w}}_t), \overline{\boldsymbol{w}}_{t+1} - \overline{\boldsymbol{w}}_t \right\rangle + \frac{L_\Phi}{2} \left\| \overline{\boldsymbol{w}}_{t+1} - \overline{\boldsymbol{w}}_t \right\|^2 \\
&= -\eta_1 \left\langle \nabla\Phi(\overline{\boldsymbol{w}}_t), \frac{1}{n}\sum_{i\in[n]} \tilde{\nabla}_{\boldsymbol{w}} f^i(w_t^i, \psi_t^i) \right\rangle + \eta_1^2 \frac{L_\Phi}{2} \left\| \frac{1}{n}\sum_{i\in[n]} \tilde{\nabla}_{\boldsymbol{w}} f^i(w_t^i, \psi_t^i) \right\|^2,
\end{aligned}
\tag{45}
$$

where we use the update rule of `FedRobust` and note that the difference of averaged models can be written as $\overline{\boldsymbol{w}}_{t+1} - \overline{\boldsymbol{w}}_t = -\eta_1 \frac{1}{n}\sum_{i\in[n]} \tilde{\nabla}_{\boldsymbol{w}} f^i(w_t^i, \psi_t^i)$. Moreover, since the stochastic gradients $\tilde{\nabla}_{\boldsymbol{w}} f^i$ are unbiased and variance-bounded by $\sigma_{\boldsymbol{w}}^2$, we can take expectation from both sides of (45) and further simplify it as follows

$$
\mathbb{E}[\Phi(\overline{\boldsymbol{w}}_{t+1})] - \mathbb{E}[\Phi(\overline{\boldsymbol{w}}_t)] \le -\frac{\eta_1}{2}\mathbb{E}\left\| \nabla\Phi(\overline{\boldsymbol{w}}_t) \right\|^2 + \frac{\eta_1}{2} h_t - \frac{\eta_1}{2}\left(1 - \eta_1 L_\Phi\right) g_t + \eta_1^2 \frac{L_\Phi}{2}\frac{\sigma_{\boldsymbol{w}}^2}{n}.
\tag{46}
$$

In above, we used the inequality $2\langle \mathbf{a}, \mathbf{b} \rangle = \|\mathbf{a}\|^2 + \|\mathbf{b}\|^2 - \|\mathbf{a} - \mathbf{b}\|^2$ as well as the notations for $g_t$ and $h_t$ as defined in Table 1.

### E.4 Proof of Lemma 4

We begin bounding $h_t$ by adding/subtracting the term $\nabla_{\boldsymbol{w}} f(\overline{\boldsymbol{w}}_t, \Psi_t)$ and use the inequality $\|\mathbf{a}+\mathbf{b}\|^2 \le 2\|\mathbf{a}\|^2 + 2\|\mathbf{b}\|^2$ to write

$$
\begin{aligned}
h_t &= \mathbb{E}\left\| \nabla\Phi(\overline{\boldsymbol{w}}_t) - \frac{1}{n}\sum_{i\in[n]} \nabla_{\boldsymbol{w}} f^i(\boldsymbol{w}_t^i, \psi_t^i) \right\|^2 \\
&\le 2\mathbb{E}\left\| \nabla\Phi(\overline{\boldsymbol{w}}_t) - \nabla_{\boldsymbol{w}} f(\overline{\boldsymbol{w}}_t, \Psi_t) \right\|^2 + 2\mathbb{E}\left\| \nabla_{\boldsymbol{w}} f(\overline{\boldsymbol{w}}_t, \Psi_t) - \frac{1}{n}\sum_{i\in[n]} \nabla_{\boldsymbol{w}} f^i(\boldsymbol{w}_t^i, \psi_t^i) \right\|^2.
\end{aligned}
\tag{47}
$$

The first term in RHS of (47) can be bounded as follows:

$$
\begin{aligned}
\mathbb{E}\left\| \nabla\Phi(\overline{\boldsymbol{w}}_t) - \nabla_{\boldsymbol{w}} f(\overline{\boldsymbol{w}}_t, \Psi_t) \right\|^2 &= \mathbb{E}\left\| \nabla_{\boldsymbol{w}} f(\overline{\boldsymbol{w}}_t, \Psi^*(\overline{\boldsymbol{w}}_t)) - \nabla_{\boldsymbol{w}} f(\overline{\boldsymbol{w}}_t, \Psi_t) \right\|^2 \\
&\overset{(a)}{\le} \frac{L_{12}^2}{n}\mathbb{E}\left\| \Psi^*(\overline{\boldsymbol{w}}_t) - \Psi_t \right\|_F^2 \\
&\overset{(b)}{\le} \frac{2L_{12}^2}{\mu_2 n}\mathbb{E}\left[ \Phi(\overline{\boldsymbol{w}}_t) - f(\overline{\boldsymbol{w}}_t, \Psi_t) \right] \\
&\overset{(c)}{=} \frac{2L_{12}^2}{\mu_2 n} b_t.
\end{aligned}
\tag{48}
$$

In above and to derive $(a)$, we employ the result of Lemma 1 which shows that given Assumption 3, the gradient function $\nabla_{\boldsymbol{w}} f(\boldsymbol{w}, \cdot)$ is $L_{12}/\sqrt{n}$ Lipschitz. To derive $(b)$, we use Assumption 4 (ii) and lastly, $(c)$ is implied from the definition of $b_t$. The second term in RHS of (47) can be bounded by noting that the local gradients $\nabla_{\boldsymbol{w}} f^i(\cdot, \psi^i)$ are $L_1$-Lipschitz, which we can write

$$
\begin{aligned}
\mathbb{E}\left\| \nabla_{\boldsymbol{w}} f(\overline{\boldsymbol{w}}_t, \Psi_t) - \frac{1}{n}\sum_{i\in[n]} \nabla_{\boldsymbol{w}} f^i(\boldsymbol{w}_t^i, \psi_t^i) \right\|^2 &= \mathbb{E}\left\| \frac{1}{n}\sum_{i\in[n]} \nabla_{\boldsymbol{w}} f^i(\overline{\boldsymbol{w}}_t, \psi_t^i) - \frac{1}{n}\sum_{i\in[n]} \nabla_{\boldsymbol{w}} f^i(\boldsymbol{w}_t^i, \psi_t^i) \right\|^2 \\
&\le \frac{L_1^2}{n}\sum_{i\in[n]} \mathbb{E}\left\| \boldsymbol{w}_t^i - \overline{\boldsymbol{w}}_t \right\|^2 \\
&= L_1^2 e_t.
\end{aligned}
\tag{49}
$$

Finally, plugging (48) and (49) back in (47) implies the claim of the lemma, that is

$$h_t \leq \frac{4L_{12}^2}{\mu_2 n} b_t + 2L_1^2 e_t. \tag{50}$$

## E.5 Proof of Lemma 5

We begin the proof by noting the definition of $b_t$ and use the fact that the gradients $\nabla_\Psi f(\boldsymbol{w}, \cdot)$ are $\frac{L_2}{n}$-Lipschitz (Refer to Lemma 1). We can accordingly write

$$\Phi(\overline{\boldsymbol{w}}_{t+1}) - f(\overline{\boldsymbol{w}}_{t+1}, \Psi_{t+1}) \leq \Phi(\overline{\boldsymbol{w}}_{t+1}) - f(\overline{\boldsymbol{w}}_{t+1}, \Psi_t) - \langle \nabla_\Psi f(\overline{\boldsymbol{w}}_{t+1}, \Psi_t), \Psi_{t+1} - \Psi_t \rangle$$
$$+ \frac{L_2}{2n} \|\Psi_{t+1} - \Psi_t\|_F^2. \tag{51}$$

In this work, we define the inner product for any two matrices $A, B$ as follows

$$\langle A, B \rangle := \mathrm{Tr}(A^\top B). \tag{52}$$

Note that according to the ascent update rule of FedRobust in Algorithm 1, we can write

$$\Psi_{t+1} - \Psi_t = \eta_2 \tilde{\partial}_t f, \tag{53}$$

where we adopt the following short-hand notation for the stochastic gradients at iteration $t$ with respect to the maximization variables $\boldsymbol{\psi}_t^i = (\Lambda_t^i, \delta_t^i)$

$$\tilde{\partial}_t f = \begin{pmatrix} \tilde{\nabla}_\psi f^1(\boldsymbol{w}_t^1, \boldsymbol{\psi}_t^1) \\ \vdots \\ \tilde{\nabla}_\psi f^n(\boldsymbol{w}_t^n, \boldsymbol{\psi}_t^n) \end{pmatrix} = \begin{pmatrix} \tilde{\nabla}_\Lambda f^1(\boldsymbol{w}_t^1, \Lambda_t^1, \delta_t^1) & \tilde{\nabla}_\delta f^1(\boldsymbol{w}_t^1, \Lambda_t^1, \delta_t^1) \\ \vdots & \vdots \\ \tilde{\nabla}_\Lambda f^n(\boldsymbol{w}_t^n, \Lambda_t^n, \delta_t^n) & \tilde{\nabla}_\delta f^n(\boldsymbol{w}_t^n, \Lambda_t^n, \delta_t^n) \end{pmatrix}. \tag{54}$$

We also denote the gradients by $\partial_t f = \mathbb{E}[\tilde{\partial}_t f]$ where the expectation is with respect to the randomness in stochastic gradients $\tilde{\nabla}_\psi f^i$. According to Assumption 2, each of the local stochastic gradients $\tilde{\nabla}_\psi f^i(\boldsymbol{w}_t^i, \boldsymbol{\psi}_t^i)$ are variance-bounded by $\sigma_\psi^2$. Therefore, we can bound the variance of $\tilde{\partial}_t f$ as $\mathbb{E}\|\tilde{\partial}_t f - \partial_t f\|_F^2 \leq n\sigma_\psi^2$. Now, we can plug these back in (51) which implies

$$\Phi(\overline{\boldsymbol{w}}_{t+1}) - \mathbb{E}f(\overline{\boldsymbol{w}}_{t+1}, \Psi_{t+1}) \leq \Phi(\overline{\boldsymbol{w}}_{t+1}) - f(\overline{\boldsymbol{w}}_{t+1}, \Psi_t) - \eta_2 \frac{n}{2} \left\| \nabla_\Psi f(\overline{\boldsymbol{w}}_{t+1}, \Psi_t) \right\|_F^2 + \eta_2^2 \frac{L_2}{2} \sigma_\psi^2$$
$$+ \eta_2 \frac{n}{2} \left\| \nabla_\Psi f(\overline{\boldsymbol{w}}_{t+1}, \Psi_t) - \frac{1}{n} \partial_t f \right\|_F^2 - \frac{\eta_2}{2n} (1 - \eta_2 L_2) \|\partial_t f\|_F^2, \tag{55}$$

where the expectation is with respect to the randomness of the stochastic gradients $\tilde{\partial}_t f$ while conditioning on all the randomness history. Now recall from Assumption 4 (ii) that $-f(\overline{\boldsymbol{w}}_{t+1}, \cdot)$ is $\mu_2$-PL implying that $\|\nabla_\Psi f(\overline{\boldsymbol{w}}_{t+1}, \Psi_t)\|_F^2 \geq 2\mu_2(\Phi(\overline{\boldsymbol{w}}_{t+1}) - f(\overline{\boldsymbol{w}}_{t+1}, \Psi_t))$. Moreover, assume that $\eta_2 \leq 1/L_2$ to remove the last term in (55). Putting altogether implies that

$$\Phi(\overline{\boldsymbol{w}}_{t+1}) - \mathbb{E}f(\overline{\boldsymbol{w}}_{t+1}, \Psi_{t+1}) \leq (1 - \mu_2 \eta_2 n) \left( \Phi(\overline{\boldsymbol{w}}_{t+1}) - f(\overline{\boldsymbol{w}}_{t+1}, \Psi_t) \right) + \eta_2^2 \frac{L_2}{2} \sigma_\psi^2$$
$$+ \eta_2 \frac{n}{2} \left\| \nabla_\Psi f(\overline{\boldsymbol{w}}_{t+1}, \Psi_t) - \frac{1}{n} \partial_t f \right\|_F^2. \tag{56}$$

Next, we continue to bound the last term in RHS of (56). We can write

$$\left\| \nabla_\Psi f(\overline{\boldsymbol{w}}_{t+1}, \Psi_t) - \frac{1}{n} \partial_t f \right\|_F^2 = \frac{1}{n^2} \sum_{i \in [n]} \left\| \nabla_\psi f^i(\overline{\boldsymbol{w}}_{t+1}, \boldsymbol{\psi}_t^i) - \nabla_\psi f^i(\boldsymbol{w}_t^i, \boldsymbol{\psi}_t^i) \right\|_F^2$$
$$\leq \frac{L_{21}^2}{n^2} \sum_{i \in [n]} \left\| \overline{\boldsymbol{w}}_{t+1} - \boldsymbol{w}_t^i \right\|^2$$
$$\leq \frac{2L_{21}^2}{n^2} \sum_{i \in [n]} \left\| \boldsymbol{w}_t^i - \overline{\boldsymbol{w}}_t \right\|^2 + \frac{2L_{21}^2}{n} \left\| \overline{\boldsymbol{w}}_{t+1} - \overline{\boldsymbol{w}}_t \right\|^2, \tag{57}$$

where the first inequality above uses Assumption 3 on Lipschitz continuity of local gradients and the second inequality simply uses the inequality $\|\mathbf{a} + \mathbf{b}\|^2 \le 2\|\mathbf{a}\|^2 + 2\|\mathbf{b}\|^2$. Next, let us bound the term $\|\overline{\boldsymbol{w}}_{t+1} - \overline{\boldsymbol{w}}_t\|^2$ in expectation as follows. Using the descent update rule in Algorithm 1 and considering Assumption 2 on variance of the stochastic gradients $\tilde{\nabla}_{\boldsymbol{w}} f^i$ we can write

$$
\begin{aligned}
\mathbb{E}\|\overline{\boldsymbol{w}}_{t+1} - \overline{\boldsymbol{w}}_t\|^2 &= \eta_1^2 \mathbb{E} \left\| \frac{1}{n} \sum_{i \in [n]} \tilde{\nabla}_{\boldsymbol{w}} f^i(\boldsymbol{w}_t^i, \boldsymbol{\psi}_t^i) \right\|^2 \\
&\le \eta_1^2 \mathbb{E} \left\| \frac{1}{n} \sum_{i \in [n]} \nabla_{\boldsymbol{w}} f^i(\boldsymbol{w}_t^i, \boldsymbol{\psi}_t^i) \right\|^2 + \eta_1^2 \frac{\sigma_{\boldsymbol{w}}^2}{n} \\
&= \eta_1^2 g_t + \eta_1^2 \frac{\sigma_{\boldsymbol{w}}^2}{n},
\end{aligned}
\tag{58}
$$

where we use the short-hand notation of $g_t$ also listed in Table 1. Plugging (58) back in (57) and noting the notation $e_t = \frac{1}{n} \sum_{i \in [n]} \mathbb{E}\|\boldsymbol{w}_t^i - \overline{\boldsymbol{w}}_t\|^2$ implies that

$$
\mathbb{E} \left\| \nabla_\Psi f(\overline{\boldsymbol{w}}_{t+1}, \Psi_t) - \frac{1}{n} \partial_t f \right\|_F^2 \le \frac{2L_{21}^2}{n} e_t + \eta_1^2 \frac{2L_{21}^2}{n} g_t + \eta_1^2 \frac{2L_{21}^2}{n} \frac{\sigma_{\boldsymbol{w}}^2}{n}.
\tag{59}
$$

Before proceeding to bound more terms, let us recall what we have shown till this point. We plug (59) back in (56), take the expectation with respect to all the sources of randomness and use the notation $b_t = \mathbb{E}[\Phi(\overline{\boldsymbol{w}}_t) - f(\overline{\boldsymbol{w}}_t, \Psi_t)]$ to conclude

$$
\begin{aligned}
b_{t+1} &\le (1 - \mu_2 \eta_2 n) \mathbb{E} \left[ \Phi(\overline{\boldsymbol{w}}_{t+1}) - f(\overline{\boldsymbol{w}}_{t+1}, \Psi_t) \right] \\
&\quad + \eta_2 L_{21}^2 e_t + \eta_1^2 \eta_2 L_{21}^2 g_t + \eta_1^2 \eta_2 L_{21}^2 \frac{\sigma_{\boldsymbol{w}}^2}{n} + \eta_2^2 \frac{L_2}{2} \sigma_\psi^2.
\end{aligned}
\tag{60}
$$

To bound the term $\mathbb{E}\left[\Phi(\overline{\boldsymbol{w}}_{t+1}) - f(\overline{\boldsymbol{w}}_{t+1}, \Psi_t)\right]$, we can decompose it to the following three terms:

$$
\Phi(\overline{\boldsymbol{w}}_{t+1}) - f(\overline{\boldsymbol{w}}_{t+1}, \Psi_t) = \Phi(\overline{\boldsymbol{w}}_t) - f(\overline{\boldsymbol{w}}_t, \Psi_t) + f(\overline{\boldsymbol{w}}_t, \Psi_t) - f(\overline{\boldsymbol{w}}_{t+1}, \Psi_t) + \Phi(\overline{\boldsymbol{w}}_{t+1}) - \Phi(\overline{\boldsymbol{w}}_t).
\tag{61}
$$

Given the Lipschitz gradient assumption for the local functions in Assumption 3 and using Lemma 1 on Lipschitz gradient for the global function, we can write

$$
f(\overline{\boldsymbol{w}}_t, \Psi_t) - f(\overline{\boldsymbol{w}}_{t+1}, \Psi_t) \le -\langle \nabla_{\boldsymbol{w}} f(\overline{\boldsymbol{w}}_t, \Psi_t), \overline{\boldsymbol{w}}_{t+1} - \overline{\boldsymbol{w}}_t \rangle + \frac{L_1}{2} \|\overline{\boldsymbol{w}}_{t+1} - \overline{\boldsymbol{w}}_t\|^2,
\tag{62}
$$

where $\overline{\boldsymbol{w}}_{t+1} - \overline{\boldsymbol{w}}_t = -\eta_1 \frac{1}{n} \sum_{i \in [n]} \tilde{\nabla}_{\boldsymbol{w}} f^i(\boldsymbol{w}_t^i, \boldsymbol{\psi}_t^i)$. Taking expectation from both sides of (62) implies that

$$
\begin{aligned}
\mathbb{E}\left[f(\overline{\boldsymbol{w}}_t, \Psi_t) - f(\overline{\boldsymbol{w}}_{t+1}, \Psi_t)\right] &\overset{(a)}{\le} \eta_1 \mathbb{E}\|\nabla_{\boldsymbol{w}} f(\overline{\boldsymbol{w}}_t, \Psi_t) - \nabla\Phi(\overline{\boldsymbol{w}}_t)\|^2 + \eta_1 \mathbb{E}\|\nabla\Phi(\overline{\boldsymbol{w}}_t)\|^2 \\
&\quad + \left( \frac{\eta_1}{2} + \eta_1^2 \frac{L_1}{2} \right) g_t + \eta_1^2 \frac{L_1}{2} \frac{\sigma_{\boldsymbol{w}}^2}{n} \\
&\overset{(b)}{\le} \eta_1 \frac{2L_{12}^2}{\mu_2 n} b_t + \eta_1 \mathbb{E}\|\nabla\Phi(\overline{\boldsymbol{w}}_t)\|^2 + \left( \frac{\eta_1}{2} + \eta_1^2 \frac{L_1}{2} \right) g_t + \eta_1^2 \frac{L_1}{2} \frac{\sigma_{\boldsymbol{w}}^2}{n},
\end{aligned}
\tag{63}
$$

where in inequality $(a)$ we use the inequality $2\langle \mathbf{a}, \mathbf{b} \rangle \le \|\mathbf{a}\|^2 + \|\mathbf{b}\|^2$ and also the result in (58). To derive $(b)$, we use Assumptions 3 and 4 (ii), result of Lemma 1 and the notation $b_t = \mathbb{E}[\Phi(\overline{\boldsymbol{w}}_t) - f(\overline{\boldsymbol{w}}_t, \Psi_t)]$ to write

$$
\begin{aligned}
\mathbb{E}\|\nabla\Phi(\overline{\boldsymbol{w}}_t) - \nabla_{\boldsymbol{w}} f(\overline{\boldsymbol{w}}_t, \Psi_t)\|^2 &= \mathbb{E}\|\nabla_{\boldsymbol{w}} f(\overline{\boldsymbol{w}}_t, \Psi^*(\overline{\boldsymbol{w}}_t)) - \nabla_{\boldsymbol{w}} f(\overline{\boldsymbol{w}}_t, \Psi_t)\|^2 \\
&\le \frac{L_{12}^2}{n} \mathbb{E}\|\Psi^*(\overline{\boldsymbol{w}}_t) - \Psi_t\|_F^2 \\
&\le \frac{2L_{12}^2}{\mu_2 n} \mathbb{E}\left[\Phi(\overline{\boldsymbol{w}}_t) - f(\overline{\boldsymbol{w}}_t, \Psi_t)\right] \\
&= \frac{2L_{12}^2}{\mu_2 n} b_t.
\end{aligned}
\tag{64}
$$

We now have all the ingredients to conclude the claim of Lemma 5. To do so, we combine the result of Lemma 3 which bounds the term $\mathbb{E}[\Phi(\overline{w}_{t+1})] - \mathbb{E}[\Phi(\overline{w}_t)]$, Lemma 4 that shows $h_t \leq 4L_{12}^2 b_t/(\mu_2 n) + 2L_1^2 e_t$, and the bound (63); plug back in (61) and then in (60) and conclude the claim of the lemma, that is

$$
b_{t+1} \leq (1 - \mu_2 \eta_2 n)\left(1 + \eta_1 \frac{4L_{12}^2}{\mu_2 n}\right) b_t + \frac{\eta_1}{2} \mathbb{E}\left\|\nabla\Phi(\overline{w}_t)\right\|^2 + \frac{\eta_1^2}{2}\left(L_1 + L_\Phi + 2\eta_2 L_{21}^2\right) g_t
$$

$$
+ \left(\eta_1 L_1^2 + \eta_2 L_{21}^2\right) e_t + \frac{\eta_1^2}{2}\left(L_1 + L_\Phi + 2\eta_2 L_{21}^2\right)\frac{\sigma_w^2}{n} + \frac{\eta_2^2}{2} L_2 \sigma_\psi^2, \tag{65}
$$

### E.6 Proof of Lemma 6

To prove this lemma, we first need to establish an intermediate step, which is stated in the following.

**Proposition 1.** *If Assumptions 1, 2 and 3 hold, then*

$$
e_t \leq 16\eta_1^2(\tau-1)L_1^2 \sum_{l=t_c+1}^{t-1} e_l + 10\eta_1^2(\tau-1)\sum_{l=t_c+1}^{t-1} g_l + 8\eta_1^2(\tau-1)^2\rho^2 + 4\eta_1^2(\tau-1)(n+1)\frac{\sigma_w^2}{n}. \tag{66}
$$

*Proof of Proposition 1.* Consider an iteration $t \geq 1$ and let $t_c$ denote the index of the most recent communication between the workers and the server, i.e. $t_c = \left\lfloor \frac{t}{\tau} \right\rfloor \tau$. Therefore, all the workers share the same local minimization model at iteration $t_c + 1$, i.e. $w_{t_c+1}^1 = \cdots = w_{t_c+1}^n = \overline{w}_{t_c+1}$. According to the update rule of FedRobust, we can write for each node $i$ that

$$
w_{t_c+2}^i = w_{t_c+1}^i - \eta_1 \tilde{\nabla}_w f^i(w_{t_c+1}^i, \psi_{t_c+1}^i),
$$
$$
\vdots
$$
$$
w_t^i = w_{t-1}^i - \eta_1 \tilde{\nabla}_w f^i(w_{t-1}^i, \psi_{t-1}^i). \tag{67}
$$

Summing up all the equalities in (67) yields that

$$
w_t^i = w_{t_c+1}^i - \eta_1 \sum_{l=t_c+1}^{t-1} \tilde{\nabla}_w f^i(w_l^i, \psi_l^i). \tag{68}
$$

Therefore, the difference of the local models $w_t^i$ and their average $\overline{w}_t$ can be written as

$$
w_t^i - \overline{w}_t = w_{t_c+1}^i - \eta_1 \sum_{l=t_c+1}^{t-1} \tilde{\nabla}_w f^i(w_l^i, \psi_l^i) - \left(\overline{w}_{t_c+1} - \eta_1 \frac{1}{n}\sum_{j\in[n]}\sum_{l=t_c+1}^{t-1} \tilde{\nabla}_w f^j(w_l^j, \psi_l^j)\right)
$$
$$
= -\eta_1\left(\sum_{l=t_c+1}^{t-1} \tilde{\nabla}_w f^i(w_l^i, \psi_l^i) - \frac{1}{n}\sum_{j\in[n]}\sum_{l=t_c+1}^{t-1} \tilde{\nabla}_w f^j(w_l^j, \psi_l^j)\right). \tag{69}
$$

This yields the following bound on each local deviation from the average $\mathbb{E}\|w_t^i - \overline{w}_t\|^2$:

$$
\mathbb{E}\left\|w_t^i - \overline{w}_t\right\|^2 = \eta_1^2 \mathbb{E}\left\|\sum_{l=t_c+1}^{t-1} \tilde{\nabla}_w f^i(w_l^i, \psi_l^i) - \frac{1}{n}\sum_{j\in[n]}\sum_{l=t_c+1}^{t-1} \tilde{\nabla}_w f^j(w_l^j, \psi_l^j)\right\|^2
$$

$$
\leq 2\eta_1^2 \mathbb{E}\left\|\sum_{l=t_c+1}^{t-1} \tilde{\nabla}_w f^i(w_l^i, \psi_l^i)\right\|^2 + 2\eta_1^2 \mathbb{E}\left\|\frac{1}{n}\sum_{j\in[n]}\sum_{l=t_c+1}^{t-1} \tilde{\nabla}_w f^j(w_l^j, \psi_l^j)\right\|^2
$$

$$
\overset{(a)}{\leq} 2\eta_1^2 \underbrace{\mathbb{E}\left\|\sum_{l=t_c+1}^{t-1} \nabla_w f^i(w_l^i, \psi_l^i)\right\|^2}_{T_3} + 2\eta_1^2 \underbrace{\mathbb{E}\left\|\frac{1}{n}\sum_{j\in[n]}\sum_{l=t_c+1}^{t-1} \nabla_w f^j(w_l^j, \psi_l^j)\right\|^2}_{T_4}
$$

$$
+ 2\eta_1^2(t - t_c - 1)(n+1)\frac{\sigma_w^2}{n}, \tag{70}
$$

where we used Assumption 2 to bound the variance of the stochastic gradients and derive $(a)$. The term $T_4$ in (70) can simply be bounded as

$$T_4 \leq \mathbb{E}\left\| \frac{1}{n} \sum_{j\in[n]} \sum_{l=t_c+1}^{t-1} \nabla_{\boldsymbol{w}} f^j(\boldsymbol{w}_l^j, \boldsymbol{\psi}_l^j) \right\|^2 \leq (t - t_c - 1) \sum_{l=t_c+1}^{t-1} \mathbb{E}\left\| \frac{1}{n} \sum_{j\in[n]} \nabla_{\boldsymbol{w}} f^j(\boldsymbol{w}_l^j, \boldsymbol{\psi}_l^j) \right\|^2 \quad (71)$$

Note that $t_c$ denotes the latest server-worker communication before iteration $t$, hence $t - t_c \leq \tau$ where $\tau$ is the duration of local updates in each round. Therefore, we have

$$T_4 \leq (\tau - 1) \sum_{l=t_c+1}^{t-1} \mathbb{E}\left\| \frac{1}{n} \sum_{j\in[n]} \nabla_{\boldsymbol{w}} f^j(\boldsymbol{w}_l^j, \boldsymbol{\psi}_l^j) \right\|^2 \leq (\tau - 1) \sum_{l=t_c+1}^{t-1} g_l \quad (72)$$

Now we proceed to bound the term $T_3$ in (70) as follows:

$$\begin{aligned}
T_3 &= \mathbb{E}\left\| \sum_{l=t_c+1}^{t-1} \nabla_{\boldsymbol{w}} f^i(\boldsymbol{w}_l^i, \boldsymbol{\psi}_l^i) \right\|^2 \\
&\leq (\tau - 1) \sum_{l=t_c+1}^{t-1} \mathbb{E}\left\| \nabla_{\boldsymbol{w}} f^i(\boldsymbol{w}_l^i, \boldsymbol{\psi}_l^i) \right\|^2 \\
&\leq 4(\tau - 1) \sum_{l=t_c+1}^{t-1} \mathbb{E}\left\| \nabla_{\boldsymbol{w}} f^i(\boldsymbol{w}_l^i, \boldsymbol{\psi}_l^i) - \nabla_{\boldsymbol{w}} f^i(\overline{\boldsymbol{w}}_l, \boldsymbol{\psi}_l^i) \right\|^2 \\
&\quad + 4(\tau - 1) \sum_{l=t_c+1}^{t-1} \mathbb{E}\left\| \nabla_{\boldsymbol{w}} f^i(\overline{\boldsymbol{w}}_l, \boldsymbol{\psi}_l^i) - \frac{1}{n} \sum_{j\in[n]} \nabla_{\boldsymbol{w}} f^j(\overline{\boldsymbol{w}}_l, \boldsymbol{\psi}_l^j) \right\|^2 \\
&\quad + 4(\tau - 1) \sum_{l=t_c+1}^{t-1} \mathbb{E}\left\| \frac{1}{n} \sum_{j\in[n]} \nabla_{\boldsymbol{w}} f^j(\overline{\boldsymbol{w}}_l, \boldsymbol{\psi}_l^j) - \frac{1}{n} \sum_{j\in[n]} \nabla_{\boldsymbol{w}} f^j(\boldsymbol{w}_l^j, \boldsymbol{\psi}_l^j) \right\|^2 \\
&\quad + 4(\tau - 1) \sum_{l=t_c+1}^{t-1} \mathbb{E}\left\| \frac{1}{n} \sum_{j\in[n]} \nabla_{\boldsymbol{w}} f^j(\boldsymbol{w}_l^j, \boldsymbol{\psi}_l^j) \right\|^2 \quad (73)
\end{aligned}$$

We can simply this bound by using Assumption 3 on Lipschitz gradients for the local objectives $f^i$s and applying the notations for $e_l$ and $g_l$ to derive

$$\begin{aligned}
T_3 &\leq 4(\tau - 1)L_1^2 \sum_{l=t_c+1}^{t-1} \mathbb{E}\left\| \boldsymbol{w}_l^i - \overline{\boldsymbol{w}}_l \right\|^2 + 4(\tau - 1) \sum_{l=t_c+1}^{t-1} \mathbb{E}\left\| \nabla_{\boldsymbol{w}} f^i(\overline{\boldsymbol{w}}_l, \boldsymbol{\psi}_l^i) - \nabla_{\boldsymbol{w}} f(\overline{\boldsymbol{w}}_l, \Psi_l) \right\|^2 \\
&\quad + 4(\tau - 1)L_1^2 \sum_{l=t_c+1}^{t-1} e_l + 4(\tau - 1) \sum_{l=t_c+1}^{t-1} g_l \quad (74)
\end{aligned}$$

We can plug (72) and (74) into (70) and take the average of the both sides over $i = 1, \cdots, n$. This implies that

$$e_t \leq 16\eta_1^2(\tau - 1)L_1^2 \sum_{l=t_c+1}^{t-1} e_l + 10\eta_1^2(\tau - 1) \sum_{l=t_c+1}^{t-1} g_l + 8\eta_1^2(\tau - 1)^2 \rho^2 + 4\eta_1^2(\tau - 1)(n + 1)\frac{\sigma_{\boldsymbol{w}}^2}{n}. \quad (75)$$

In above, we used the result of Proposition 2 that given Assumption 1, bounds the gradient diversity $\frac{1}{n} \sum_{i\in[n]} \|\nabla_{\boldsymbol{w}} f^i(\boldsymbol{w}, \boldsymbol{\psi}^i) - \nabla_{\boldsymbol{w}} f(\boldsymbol{w}, \Psi)\|^2 \leq \rho^2$, where $\rho^2 = 3\rho_f^2 + 6L_{12}^2(\epsilon_1^2 + \epsilon_2^2)$. We defer the proof this proposition to the end of this section. This concludes the proof of Proposition 1. $\square$

Having set the required intermediate steps, we resume the proof of Lemma 6. According to Proposition 1, we can write the term $e_t$ as follows

$$e_t \leq C_1 \sum_{l=t_c+1}^{t-1} e_l + C_2 \sum_{l=t_c+1}^{t-1} g_l + C_3 \quad (76)$$

where we use the following short-hand coefficients

$$C_1 \coloneqq 16\eta_1^2(\tau-1)L_1^2$$
$$C_2 \coloneqq 10\eta_1^2(\tau-1)$$
$$C_3 \coloneqq 8\eta_1^2(\tau-1)^2\rho^2 + 4\eta_1^2(\tau-1)(n+1)\frac{\sigma_{\boldsymbol{w}}^2}{n}. \tag{77}$$

We can then write this bound for every iteration in $[t_c+1:t]$, that is

$$e_{t_c+1} = 0$$
$$e_{t_c+2} \le C_1 e_{t_c+1} + C_2 g_{t_c+1} + C_3$$
$$\vdots$$
$$e_t \le C_1\left(e_{t_c+1} + \cdots + e_{t-1}\right) + C_2\left(g_{t_c+1} + \cdots + g_{t-1}\right) + C_3. \tag{78}$$

Summing all of the inequalities results in the following

$$\sum_{l=t_c+1}^{t-1} e_l \le C_1(\tau-1)\sum_{l=t_c+1}^{t-1} e_l + C_2(\tau-1)\sum_{l=t_c+1}^{t-1} g_l + C_3(\tau-1). \tag{79}$$

We can further rearrange the terms above and write

$$\sum_{l=t_c+1}^{t-1} e_l \le \frac{C_2(\tau-1)}{1-C_1(\tau-1)}\sum_{l=t_c+1}^{t-1} g_l + \frac{C_3(\tau-1)}{1-C_1(\tau-1)}. \tag{80}$$

Now, if we assume that $C_1(\tau-1) \le 1/2$, then we get the following bound on $\sum_{l=t_c+1}^{t-1} e_l$

$$\sum_{l=t_c+1}^{t-1} e_l \le 2C_2(\tau-1)\sum_{l=t_c+1}^{t-1} g_l + 2C_3(\tau-1) \tag{81}$$

Plugging back in (99) and using the assumption $C_1(\tau-1) \le 1/2$ yields that

$$e_t \le C_1\left(2C_2(\tau-1)\sum_{l=t_c+1}^{t-1} g_l + 2C_3(\tau-1)\right) + C_2\sum_{l=t_c+1}^{t-1} g_l + C_3$$
$$\le 2C_2\sum_{l=t_c+1}^{t-1} g_l + 2C_3, \tag{82}$$

which concludes the proof of Lemma 6. Lastly, we present the following proposition along with its proof which we used this result to prove Proposition 1.

**Proposition 2.** *An immediate implication of Assumptions 1 and 3 is that for any $\boldsymbol{w}, \Psi$, the diversity of the local gradients is bounded in the following sense*

$$\frac{1}{n}\sum_{i\in[n]}\left\|\nabla_{\boldsymbol{w}}f^i(\boldsymbol{w},\boldsymbol{\psi}^i) - \nabla_{\boldsymbol{w}}f(\boldsymbol{w},\Psi)\right\|^2 \le \rho^2, \tag{83}$$

*where we denote $\rho^2 = 3\rho_f^2 + 6L_{12}^2(\epsilon_1^2 + \epsilon_2^2)$.*

*Proof of Proposition 2.* The proof is simply implied from Assumptions 1 and 3 by writing

$$\frac{1}{n}\sum_{i\in[n]}\left\|\nabla_{\boldsymbol{w}}f^i(\boldsymbol{w},\boldsymbol{\psi}^i) - \nabla_{\boldsymbol{w}}f(\boldsymbol{w},\Psi)\right\|^2 \le 3\frac{1}{n}\sum_{i\in[n]}\left\|\nabla_{\boldsymbol{w}}f^i(\boldsymbol{w},\Lambda^i,\delta^i) - \nabla_{\boldsymbol{w}}f^i(\boldsymbol{w},I,0)\right\|^2$$
$$+ 3\frac{1}{n}\sum_{i\in[n]}\left\|\nabla_{\boldsymbol{w}}f^i(\boldsymbol{w}) - \nabla_{\boldsymbol{w}}f(\boldsymbol{w})\right\|^2$$
$$+ 3\frac{1}{n}\sum_{i\in[n]}\left\|\nabla_{\boldsymbol{w}}f(\boldsymbol{w},I,0) - \nabla_{\boldsymbol{w}}f(\boldsymbol{w},\Psi)\right\|^2$$
$$\le 3\rho_f^2 + 6L_{12}^2(\epsilon_1^2 + \epsilon_2^2). \tag{84}$$

$\square$

## E.7 Proof of Lemma 7

[16] proves a similar claim for $\Gamma = 0$. For completeness, we provide the proof for general case when $\Gamma \neq 0$. Let $t_c$ denote the index of the most recent communication round, i.e. $t_c = \lfloor \frac{t}{\tau} \rfloor \tau$. We can write $t = t_c + r$ where $1 \leq r \leq \tau$. Starting from $r = 1$, we can write

$$
\begin{aligned}
P_{t_c+2} &\leq \Upsilon P_{t_c+1} - \frac{\eta_1}{2} \left(1 - \eta_1 L\right) g_{t_c+1} + \Gamma \\
&\leq \Upsilon P_{t_c+1} + \Gamma,
\end{aligned}
\tag{85}
$$

where the last inequality holds if

$$
\eta_1 L \leq 1. \tag{86}
$$

We can continue for $r = 2$ as follows

$$
\begin{aligned}
P_{t_c+3} &\leq \Upsilon P_{t_c+2} - \frac{\eta_1}{2} \left(1 - \eta_1 L\right) g_{t_c+2} + \eta_1^2 B g_{t_c+1} + \Gamma \\
&\overset{(a)}{\leq} \Upsilon^2 P_{t_c+1} - \frac{\eta_1}{2} \Upsilon \left(1 - \eta_1 L - \eta_1 \frac{2B}{\Upsilon}\right) g_{t_c+1} + \Gamma(1 + \Upsilon) \\
&\overset{(b)}{\leq} \Upsilon^2 P_{t_c+1} + \Gamma(1 + \Upsilon)
\end{aligned}
\tag{87}
$$

where $(a)$ is due to the inequality $P_{t_c+2} \leq \Upsilon P_{t_c+1} - \frac{\eta_1}{2}(1 - \eta_1 L)g_{t_c+1} + \Gamma$ and $(b)$ holds if

$$
1 - \eta_1 L - \eta_1 \frac{2B}{\Upsilon} \geq 0, \tag{88}
$$

or equivalently

$$
\eta_1 \left(L + \frac{2B}{\Upsilon}\right) \leq 1. \tag{89}
$$

We can continue the same argument up to $r + 1$ and write

$$
P_{t_c+r+1} \leq \Upsilon^r P_{t_c+1} + \Gamma(1 + \Upsilon + \cdots + \Upsilon^{r-1}), \tag{90}
$$

if the step-size is as small as follows

$$
\eta_1 \left(L + \frac{2B}{\Upsilon^{r-1}} \left(1 + \Upsilon + \cdots + \Upsilon^{r-2}\right)\right) \leq 1. \tag{91}
$$

Since $1 + \Upsilon + \cdots + \Upsilon^{r-2} \leq \frac{1}{1-\Upsilon}$, then the following condition implies all the previous ones on $\eta$

$$
\eta_1 \left(L + \frac{2B}{\Upsilon^{r-1}(1 - \Upsilon)}\right). \tag{92}
$$

Moreover, since $\Upsilon < 1$, then the strongest condition on $\eta$ is (92) when we put the largest possible value for $r$ which is $\tau$, yielding

$$
\eta_1 \left(L + \frac{2B}{\Upsilon^{\tau-1}(1 - \Upsilon)}\right). \tag{93}
$$

Lastly, we note that $1 + \Upsilon + \cdots + \Upsilon^{r-1} \leq \frac{1}{1-\Upsilon}$ in (90), and the claim is concluded.

## E.8 Proof of Lemma 8

Recall the result of Lemma 5 in which we showed that if $\eta_2 \leq 1/L_2$, then the following contraction bound on the sequence $\{b_t\}_{t \geq 0}$ holds:

$$
\begin{aligned}
b_{t+1} &\leq (1 - \mu_2 \eta_2 n) \left(1 + \eta_1 \frac{4L_{12}^2}{\mu_2 n}\right) b_t + \frac{\eta_1}{2} \mathbb{E}\|\nabla \Phi(\overline{\boldsymbol{w}}_t)\|^2 + \frac{\eta_1^2}{2} \left(L_1 + L_\Phi + 2\eta_2 L_{21}^2\right) g_t \\
&\quad + \left(\eta_1 L_1^2 + \eta_2 L_{21}^2\right) e_t + \frac{\eta_1^2}{2} \left(L_1 + L_\Phi + 2\eta_2 L_{21}^2\right) \frac{\sigma_{\boldsymbol{w}}^2}{n} + \frac{\eta_2^2}{2} L_2 \sigma_\psi^2,
\end{aligned}
\tag{94}
$$

and consider the coefficient of $b_t$ in above. A simple calculation yields that if the step-sizes satisfy the condition $\frac{\eta_2}{\eta_1} \geq \frac{8L_{12}^2}{\mu_2^2 n^2}$, then we have

$$(1 - \mu_2\eta_2 n)\left(1 + \eta_1 \frac{4L_{12}^2}{\mu_2 n}\right) \leq 1 - \frac{1}{2}\mu_2\eta_2 n. \tag{95}$$

Now, we denote $\gamma = 1 - \frac{1}{2}\mu_2\eta_2 n$ and apply (94) to all iterations $t = 0, \cdots, T-1$, which yields that

$$b_0 \leq \frac{2L_2^2}{\mu_2 n}\left(\epsilon_1^2 + \epsilon_2^2\right),$$

$$b_1 \leq \gamma b_0 + \frac{\eta_1}{2}\mathbb{E}\left\|\nabla\Phi(\overline{\boldsymbol{w}}_t)\right\|^2 + \frac{\eta_1^2}{2}\left(L_1 + L_\Phi + 2\eta_2 L_{21}^2\right)g_0 + \left(\eta_1 L_1^2 + \eta_2 L_{21}^2\right)e_0$$

$$+ \frac{\eta_1^2}{2}\left(L_1 + L_\Phi + 2\eta_2 L_{21}^2\right)\frac{\sigma_{\boldsymbol{w}}^2}{n} + \frac{\eta_2^2}{2}L_2\sigma_\psi^2,$$

$$\vdots$$

$$b_{T-1} \leq \gamma b_{T-2} + \frac{\eta_1}{2}\mathbb{E}\left\|\nabla\Phi(\overline{\boldsymbol{w}}_t)\right\|^2 + \frac{\eta_1^2}{2}\left(L_1 + L_\Phi + 2\eta_2 L_{21}^2\right)g_{T-2} + \left(\eta_1 L_1^2 + \eta_2 L_{21}^2\right)e_{T-2}$$

$$+ \frac{\eta_1^2}{2}\left(L_1 + L_\Phi + 2\eta_2 L_{21}^2\right)\frac{\sigma_{\boldsymbol{w}}^2}{n} + \frac{\eta_2^2}{2}L_2\sigma_\psi^2. \tag{96}$$

Taking the average of the $T$ inequalities above yields that

$$(1-\gamma)\frac{1}{T}\sum_{t=0}^{T-1} b_t \leq \frac{2L_2^2}{\mu_2 n}\frac{\epsilon_1^2 + \epsilon_2^2}{T} + \frac{\eta_1}{2}\frac{1}{T}\sum_{t=0}^{T-1}\mathbb{E}\left\|\nabla\Phi(\overline{\boldsymbol{w}}_t)\right\|^2$$

$$+ \frac{\eta_1^2}{2}\left(L_1 + L_\Phi + 2\eta_2 L_{21}^2\right)\frac{1}{T}\sum_{t=0}^{T-1} g_t + \left(\eta_1 L_1^2 + \eta_2 L_{21}^2\right)\frac{1}{T}\sum_{t=0}^{T-1} e_t$$

$$+ \frac{\eta_1^2}{2}\left(L_1 + L_\Phi + 2\eta_2 L_{21}^2\right)\frac{\sigma_{\boldsymbol{w}}^2}{n} + \frac{\eta_2^2}{2}L_2\sigma_\psi^2. \tag{97}$$

We can further divide both sides of (97) by $1 - \gamma$ and conclude

$$\frac{1}{T}\sum_{t=0}^{T-1} b_t \leq \frac{4L_2^2}{\mu_2^2 n^2}\frac{\epsilon_1^2 + \epsilon_2^2}{\eta_2 T} + \frac{\eta_1}{\eta_2}\frac{1}{\mu_2 n}\frac{1}{T}\sum_{t=0}^{T-1}\mathbb{E}\left\|\nabla\Phi(\overline{\boldsymbol{w}}_t)\right\|^2$$

$$+ \frac{\eta_1^2}{\eta_2}\frac{1}{\mu_2 n}\left(L_1 + L_\Phi + 2\eta_2 L_{21}^2\right)\frac{1}{T}\sum_{t=0}^{T-1} g_t + \frac{1}{\eta_2}\frac{2}{\mu_2 n}\left(\eta_1 L_1^2 + \eta_2 L_{21}^2\right)\frac{1}{T}\sum_{t=0}^{T-1} e_t$$

$$+ \frac{\eta_1^2}{\eta_2}\frac{1}{\mu_2 n}\left(L_1 + L_\Phi + 2\eta_2 L_{21}^2\right)\frac{\sigma_{\boldsymbol{w}}^2}{n} + \eta_2\frac{L_2}{\mu_2 n}\sigma_\psi^2. \tag{98}$$

### E.9 Proof of Lemma 9

We begin by noting the result of Proposition 1 in which we showed the following bound on $e_t$

$$e_t \leq C_1\sum_{l=t_c+1}^{t-1} e_l + C_2\sum_{l=t_c+1}^{t-1} g_l + C_3, \tag{99}$$

where we defined the coefficients $C_1, C_2, C_3$ in (77) and recall here for more convenient:

$$C_1 := 16\eta_1^2(\tau-1)L_1^2$$

$$C_2 := 10\eta_1^2(\tau-1)$$

$$C_3 := 8\eta_1^2(\tau-1)^2\rho^2 + 4\eta_1^2(\tau-1)(n+1)\frac{\sigma_{\boldsymbol{w}}^2}{n}. \tag{100}$$

Next, we apply this bound to each iteration $t = 0, \cdots, T-1$ as follows

$$e_0 = 0$$

$$
\begin{cases}
e_1 & = 0 \\
e_2 & \leq C_1 e_1 + C_2 g_1 + C_3 \\
\vdots \\
e_\tau & \leq C_1 \left(e_1 + \cdots + e_{\tau-1}\right) + C_2 \left(g_1 + \cdots + g_{\tau-1}\right) + C_3
\end{cases}
$$

$$
\begin{cases}
e_{\tau+1} & = 0 \\
e_{\tau+2} & \leq C_1 e_{\tau+1} + C_2 g_{\tau+1} + C_3 \\
\vdots \\
e_{2\tau} & \leq C_1 \left(e_{\tau+1} + \cdots + e_{2\tau-1}\right) + C_2 \left(g_{\tau+1} + \cdots + g_{2\tau-1}\right) + C_3
\end{cases}
$$

$$\vdots$$

$$
\begin{cases}
e_{T_c+1} & = 0 \\
e_{T_c+2} & \leq C_1 e_{T_c+1} + C_2 g_{T_c+1} + C_3 \\
\vdots \\
e_{T-1} & \leq C_1 \left(e_{T_c+1} + \cdots + e_{T-2}\right) + C_2 \left(g_{T_c+1} + \cdots + g_{T-2}\right) + C_3,
\end{cases}
\tag{101}
$$

where $T_c = \lfloor \frac{T}{\tau} \rfloor \tau$ denote the index of the most recent communication between the workers and the server before iteration $T$. Summing the above inequalities yields that

$$\sum_{t=0}^{T-1} e_t \leq C_1(\tau-1) \sum_{t=0}^{T-1} e_t + C_2(\tau-1) \sum_{t=0}^{T-1} g_t + C_3 T. \tag{102}$$

Now if we assume that $C_1(\tau-1) = 16\eta_1^2(\tau-1)^2 L_1^2 \leq \frac{1}{2}$, the the claim is concluded by rearranging the terms in (102):

$$\frac{1}{T} \sum_{t=0}^{T-1} e_t \leq 2C_2(\tau-1) \frac{1}{T} \sum_{t=0}^{T-1} g_t + 2C_3. \tag{103}$$

## Appendix F   Proof of Theorem 3

Fix a distribution $\tilde{P}$ and consider

$$\max_{\Lambda, \delta} \mathbb{E}_{\tilde{P}}[\ell(f_{\boldsymbol{w}}(\Lambda \mathbf{x} + \delta))] - \lambda \|\delta\|_2^2 - \lambda \|\Lambda - I\|_F^2 \tag{104}$$

Assuming a 1-Lipschitz loss $\ell$ with 1-Lipschitz gradient, based on [36]'s Lemma 7 the above function's gradient with respect to $\delta$ has a Lipschitz constant bounded by

$$\mathrm{Lip}(\nabla f_{\boldsymbol{w}}) := \left(\prod_{i=1}^{L} \|\boldsymbol{w}_i\|_\sigma\right) \sum_{i=1}^{l} \prod_{j=1}^{i} \|\boldsymbol{w}_j\|_\sigma.$$

Similarly, the expected loss's derivative with respect to $\Lambda$ will also be Lipschitz in the spectral norm with a Lipschitz constant upper-bounded by

$$B \, \mathrm{Lip}(\nabla f_{\boldsymbol{w}}) = B \left(\prod_{i=1}^{L} \|\boldsymbol{w}_i\|_\sigma\right) \sum_{i=1}^{l} \prod_{j=1}^{i} \|\boldsymbol{w}_j\|_\sigma.$$

Given weights in $\boldsymbol{w}$, we denote the optimal solution for $\delta$ and $\Lambda$ by $\delta_{\boldsymbol{w}}$ and $\Lambda_{\boldsymbol{w}}$, respectively. To apply the Pac-Bayes generalization analysis, we need to bound the change in $\delta_{\boldsymbol{w}}, \Lambda_{\boldsymbol{w}}$ caused by perturbing $\boldsymbol{w}$ to $\boldsymbol{w} + \boldsymbol{u}$. Note that since $\lambda > (1+B) \, \mathrm{Lip}(\nabla f_{\boldsymbol{w}})$, the maximization problem for optimizing $\Lambda_{\boldsymbol{w}}, \delta_{\boldsymbol{w}}$ is maximizing a strongly-concave objective whose solutions will satisfy:

$$\delta_{\boldsymbol{w}} = \frac{1}{\lambda} \mathbb{E}[\nabla \ell \circ f_{\boldsymbol{w}}(\Lambda_{\boldsymbol{w}} \mathbf{x} + \delta_{\boldsymbol{w}})],$$

$$\Lambda_{\boldsymbol{w}} - I = \frac{1}{\lambda} \mathbb{E}[(\nabla \ell \circ f_{\boldsymbol{w}}(\Lambda_{\boldsymbol{w}} \mathbf{x} + \delta_{\boldsymbol{w}})) \mathbf{X}^\top]$$

which are norm-bounded by $\frac{\text{Lip}(\ell \circ f_{\boldsymbol{w}})}{\lambda} \leq \frac{\Pi_{i=1}^{d}\|\boldsymbol{w}_i\|_\sigma}{\lambda}$ and $B\frac{\text{Lip}(\ell \circ f_{\boldsymbol{w}})}{\lambda} \leq B\frac{\Pi_{i=1}^{d}\|\boldsymbol{w}_i\|_\sigma}{\lambda}$, respectively. Therefore, for a norm-bounded perturbation $\boldsymbol{u}$ where $\|\boldsymbol{u}_i\|_\sigma \leq \frac{1}{L}\|\boldsymbol{w}_i\|_\sigma$ we can write

$$
\begin{aligned}
&\left\|\delta_{\boldsymbol{w}+\boldsymbol{u}} - \delta_{\boldsymbol{w}}\right\|_2 + \left\|\Lambda_{\boldsymbol{w}+\boldsymbol{u}} - \Lambda_{\boldsymbol{w}}\right\|_\sigma \\
&= \left\|\frac{1}{\lambda}\mathbb{E}[\nabla\ell(f_{\boldsymbol{w}+\boldsymbol{u}}(\Lambda_{\boldsymbol{w}+\boldsymbol{u}}\mathbf{X} + \delta_{\boldsymbol{w}+\boldsymbol{u}}))] - \frac{1}{\lambda}\mathbb{E}[\nabla\ell(f_{\boldsymbol{w}}(\Lambda_{\boldsymbol{w}}\mathbf{X} + \delta_{\boldsymbol{w}}))]\right\|_2 \\
&\quad + \left\|\frac{1}{\lambda}\mathbb{E}[\nabla\ell(f_{\boldsymbol{w}+\boldsymbol{u}}(\Lambda_{\boldsymbol{w}+\boldsymbol{u}}\mathbf{X} + \delta_{\boldsymbol{w}+\boldsymbol{u}}))\mathbf{X}^\top] - \frac{1}{\lambda}\mathbb{E}[\nabla\ell(f_{\boldsymbol{w}}(\Lambda_{\boldsymbol{w}}\mathbf{X} + \delta_{\boldsymbol{w}}))\mathbf{X}^\top]\right\|_\sigma \\
&= \left\|\frac{1}{\lambda}\mathbb{E}[\nabla\ell(f_{\boldsymbol{w}+\boldsymbol{u}}(\Lambda_{\boldsymbol{w}+\boldsymbol{u}}\mathbf{X} + \delta_{\boldsymbol{w}+\boldsymbol{u}})) - \nabla\ell(f_{\boldsymbol{w}}(\Lambda_{\boldsymbol{w}}\mathbf{X} + \delta_{\boldsymbol{w}}))]\right\|_2 \\
&\quad + \left\|\frac{1}{\lambda}\mathbb{E}[(\nabla\ell(f_{\boldsymbol{w}+\boldsymbol{u}}(\Lambda_{\boldsymbol{w}+\boldsymbol{u}}\mathbf{X} + \delta_{\boldsymbol{w}+\boldsymbol{u}})) - \nabla\ell(f_{\boldsymbol{w}}(\Lambda_{\boldsymbol{w}}\mathbf{X} + \delta_{\boldsymbol{w}})))\mathbf{X}^\top]\right\|_\sigma \\
&\leq \left\|\frac{1}{\lambda}\mathbb{E}[\nabla\ell(f_{\boldsymbol{w}+\boldsymbol{u}}(\Lambda_{\boldsymbol{w}+\boldsymbol{u}}\mathbf{X} + \delta_{\boldsymbol{w}+\boldsymbol{u}})) - \nabla\ell(f_{\boldsymbol{w}}(\Lambda_{\boldsymbol{w}+\boldsymbol{u}}\mathbf{X} + \delta_{\boldsymbol{w}+\boldsymbol{u}}))]\right\|_2 \\
&\quad + \left\|\frac{1}{\lambda}\mathbb{E}[\nabla\ell(f_{\boldsymbol{w}}(\Lambda_{\boldsymbol{w}+\boldsymbol{u}}\mathbf{X} + \delta_{\boldsymbol{w}+\boldsymbol{u}})) - \nabla\ell(f_{\boldsymbol{w}}(\Lambda_{\boldsymbol{w}}\mathbf{X} + \delta_{\boldsymbol{w}+\boldsymbol{u}}))]\right\|_2 \\
&\quad + \left\|\frac{1}{\lambda}\mathbb{E}[\nabla\ell(f_{\boldsymbol{w}}(\Lambda_{\boldsymbol{w}}\mathbf{X} + \delta_{\boldsymbol{w}+\boldsymbol{u}})) - \nabla\ell(f_{\boldsymbol{w}}(\Lambda_{\boldsymbol{w}}\mathbf{X} + \delta_{\boldsymbol{w}}))]\right\|_2 \\
&\quad + \left\|\frac{1}{\lambda}\mathbb{E}[(\nabla\ell(f_{\boldsymbol{w}+\boldsymbol{u}}(\Lambda_{\boldsymbol{w}+\boldsymbol{u}}\mathbf{X} + \delta_{\boldsymbol{w}+\boldsymbol{u}})) - \nabla\ell(f_{\boldsymbol{w}}(\Lambda_{\boldsymbol{w}+\boldsymbol{u}}\mathbf{X} + \delta_{\boldsymbol{w}+\boldsymbol{u}})))\mathbf{X}^\top]\right\|_\sigma \\
&\quad + \left\|\frac{1}{\lambda}\mathbb{E}[(\nabla\ell(f_{\boldsymbol{w}}(\Lambda_{\boldsymbol{w}+\boldsymbol{u}}\mathbf{X} + \delta_{\boldsymbol{w}+\boldsymbol{u}})) - \nabla\ell(f_{\boldsymbol{w}}(\Lambda_{\boldsymbol{w}}\mathbf{X} + \delta_{\boldsymbol{w}+\boldsymbol{u}})))\mathbf{X}^\top]\right\|_\sigma \\
&\quad + \left\|\frac{1}{\lambda}\mathbb{E}[(\nabla\ell(f_{\boldsymbol{w}}(\Lambda_{\boldsymbol{w}}\mathbf{X} + \delta_{\boldsymbol{w}+\boldsymbol{u}})) - \nabla\ell(f_{\boldsymbol{w}}(\Lambda_{\boldsymbol{w}}\mathbf{X} + \delta_{\boldsymbol{w}})))\mathbf{X}^\top]\right\|_\sigma \\
&\leq \frac{(B+1)\text{lip}(\ell \circ f_{\boldsymbol{w}})}{\lambda}\left(\|\delta_{\boldsymbol{w}+\boldsymbol{u}} - \delta_{\boldsymbol{w}}\|_2 + \|\Lambda_{\boldsymbol{w}+\boldsymbol{u}} - \Lambda_{\boldsymbol{w}}\|_\sigma\right) \\
&\quad + (B+1)e^2(\prod_{i=1}^{L}\|\boldsymbol{w}_i\|_\sigma)\sum_{i=1}^{d}\left[\frac{\|\boldsymbol{u}_i\|_\sigma}{\|\boldsymbol{w}_i\|_\sigma} + B(\prod_{j=1}^{i}\|\boldsymbol{w}_j\|_\sigma)\sum_{j=1}^{i}\frac{\|\boldsymbol{u}_j\|_\sigma}{\|\boldsymbol{w}_j\|_\sigma}\right],
\end{aligned}
$$

where the last inequality follows from Lemma 3 in [36]. As a result,

$$
\begin{aligned}
&\left\|\delta_{\boldsymbol{w}+\boldsymbol{u}} - \delta_{\boldsymbol{w}}\right\|_2 + \left\|\Lambda_{\boldsymbol{w}+\boldsymbol{u}} - \Lambda_{\boldsymbol{w}}\right\|_\sigma \\
&\leq \frac{\lambda}{\lambda - (B+1)\text{lip}(\ell \circ f_{\boldsymbol{w}})}\left[(B+1)e^2(\prod_{i=1}^{L}\|\boldsymbol{w}_i\|_\sigma)\sum_{i=1}^{d}\left[\frac{\|\boldsymbol{u}_i\|_\sigma}{\|\boldsymbol{w}_i\|_\sigma} + B(\prod_{j=1}^{i}\|\boldsymbol{w}_j\|_\sigma)\sum_{j=1}^{i}\frac{\|\boldsymbol{u}_j\|_\sigma}{\|\boldsymbol{w}_j\|_\sigma}\right]\right].
\end{aligned}
$$

Then, we can bound the change in the loss function caused by perturbing $\boldsymbol{w}$ at any $\|\mathbf{x}\|_2 \leq B$ with any norm-bounded $\|\boldsymbol{u}_i\|_\sigma \leq \frac{1}{L}\|\boldsymbol{w}_i\|_\sigma$:

$$
\begin{aligned}
&\left\|f_{\boldsymbol{w}+\boldsymbol{u}}(\Lambda_{\boldsymbol{w}+\boldsymbol{u}}\mathbf{X} + \delta_{\boldsymbol{w}+\boldsymbol{u}}) - f_{\boldsymbol{w}}(\Lambda_{\boldsymbol{w}}\mathbf{X} + \delta_{\boldsymbol{w}})\right\|_2 \\
&\leq \left\|f_{\boldsymbol{w}+\boldsymbol{u}}(\Lambda_{\boldsymbol{w}+\boldsymbol{u}}\mathbf{X} + \delta_{\boldsymbol{w}+\boldsymbol{u}}) - f_{\boldsymbol{w}}(\Lambda_{\boldsymbol{w}+\boldsymbol{u}}\mathbf{X} + \delta_{\boldsymbol{w}+\boldsymbol{u}})\right\|_2 \\
&\quad + \left\|f_{\boldsymbol{w}}(\Lambda_{\boldsymbol{w}+\boldsymbol{u}}\mathbf{X} + \delta_{\boldsymbol{w}+\boldsymbol{u}}) - f_{\boldsymbol{w}}(\Lambda_{\boldsymbol{w}}\mathbf{X} + \delta_{\boldsymbol{w}+\boldsymbol{u}})\right\|_2 \\
&\quad + \left\|f_{\boldsymbol{w}}(\Lambda_{\boldsymbol{w}}\mathbf{X} + \delta_{\boldsymbol{w}+\boldsymbol{u}}) - f_{\boldsymbol{w}}(\Lambda_{\boldsymbol{w}}\mathbf{X} + \delta_{\boldsymbol{w}})\right\|_2 \\
&\leq eB\big(\prod_{i=1}^{L}\|\boldsymbol{w}_i\|_\sigma\big)\sum_{i=1}^{L}\frac{\|\boldsymbol{u}_i\|_2}{\|\boldsymbol{w}_i\|_2} + (1+B)\big(\prod_{i=1}^{d}\|\boldsymbol{w}_i\|_\sigma\big) \\
&\quad \frac{e^2}{\lambda - (B+1)\text{Lip}(\nabla f_{\boldsymbol{w}})}\sum_{i=1}^{L}\left[\frac{\|\boldsymbol{u}_i\|_\sigma}{\|\boldsymbol{w}_i\|_\sigma} + B(\prod_{j=1}^{i}\|\boldsymbol{w}_j\|_\sigma)\sum_{j=1}^{i}\frac{\|\boldsymbol{u}_j\|_\sigma}{\|\boldsymbol{w}_j\|_\sigma}\right].
\end{aligned}
$$

Now, for a fixed weight vector $\tilde{\boldsymbol{w}}$ we consider a multivariate Gaussian distribution $Q$ with zero-mean and diagonal covaraince matrix for perturbation $\boldsymbol{u}$ where each entry $\boldsymbol{u}_i$ has standard deviation

$\kappa_i = \frac{\|\tilde{\boldsymbol{w}}_i\|_\sigma}{\sqrt[L]{\prod_{i=1}^L \|\tilde{\boldsymbol{w}}_i\|_\sigma}} \kappa$ with $\kappa$ chosen as

$$\kappa = \frac{\gamma}{8e^5 L \sqrt{2d\log(4dL)} B \left(\prod_{i=1}^L \|\tilde{\boldsymbol{w}}_i\|_\sigma\right)\left(1 + \frac{\lambda}{\lambda - (1+B)\overline{\mathrm{Lip}}(\nabla f_{\boldsymbol{w}})} \sum_{i=1}^L \prod_{j=1}^i \|\tilde{\boldsymbol{w}}_j\|_\sigma\right)}. \quad (105)$$

Also, for any $\boldsymbol{w}$ which satisfies $\|\boldsymbol{w}_i\|_\sigma - \|\tilde{\boldsymbol{w}}_i\|_\sigma | \le \frac{\eta}{4L}\|\tilde{\boldsymbol{w}}_i\|_\sigma$, we have $\overline{\mathrm{Lip}}(\ell \circ f_{\boldsymbol{w}}) \le e^{\eta/2}\lambda(1-\eta) \le (1-\eta/2)\lambda$. Therefore,

$$\begin{aligned}
&\mathrm{KL}(P_{\boldsymbol{w}+\boldsymbol{u}}\|Q) \\
&\le \sum_{i=1}^d \frac{\|\boldsymbol{w}_i\|_F^2}{2\kappa_i^2} \\
&\le O\left( L^2 B^2 d\log(dL) \frac{\left(\prod_{i=1}^L \|\tilde{\boldsymbol{w}}_i\|_\sigma^2\right)\left(1 + \frac{1}{\lambda - (1+B)\overline{\mathrm{Lip}}(\nabla f_{\boldsymbol{w}})} \sum_{i=1}^L \prod_{j=1}^i \|\tilde{\boldsymbol{w}}_j\|_\sigma\right)^2}{\gamma^2} \sum_{i=1}^d \frac{\|\boldsymbol{w}_i\|_F^2}{\|\tilde{\boldsymbol{w}}_i\|_\sigma^2} \right) \\
&\le O\left( L^2 B^2 d\log(dL) \frac{\left(\prod_{i=1}^L \|\boldsymbol{w}_i\|_\sigma^2\right)\left(1 + \frac{1}{\lambda - (1+B)\overline{\mathrm{Lip}}(\nabla f_{\boldsymbol{w}})} \sum_{i=1}^L \prod_{j=1}^i \|\tilde{\boldsymbol{w}}_j\|_\sigma\right)^2}{\gamma^2} \sum_{i=1}^d \frac{\|\boldsymbol{w}_i\|_F^2}{\|\boldsymbol{w}_i\|_\sigma^2} \right)
\end{aligned}$$

Now we plug the above result into [36]'s Lemma 1, implying that given a fixed underlying distribution $P$ and any $\xi > 0$ with probability at least $1 - \xi$ for any $\boldsymbol{w}$ satisfying $\|\boldsymbol{w}_i\|_\sigma - \|\tilde{\boldsymbol{w}}_i\|_\sigma | \le \frac{\eta}{4L}\|\tilde{\boldsymbol{w}}_i\|_\sigma$ we have

$$\mathcal{L}_{0-1}^{\mathrm{adv}}(\boldsymbol{w}) - \hat{\mathcal{L}}_\gamma^{\mathrm{adv}}(\boldsymbol{w}) \le \mathcal{O}\left( \sqrt{\frac{B^2 L^2 d\log(Ld)\lambda^2 \left(\prod_{i=1}^L \|\boldsymbol{w}_i\|_\sigma \sum_{i=1}^L \frac{\|\boldsymbol{w}_i\|_F^2}{\|\boldsymbol{w}_i\|_\sigma^2}\right)^2 + \log\frac{m}{\xi}}{m\gamma^2(\lambda - (1+B)\mathrm{Lip}(\nabla f_{\boldsymbol{w}}))^2}} \right). \quad (106)$$

Now we use a cover of size $O(\frac{L}{\eta}\log M)$ points where for any feasible $\|\boldsymbol{w}_i\|_\sigma$ we can find a point $a_i$ in the cover such that $\|\boldsymbol{w}_i\|_\sigma - a_i | \le \frac{\eta}{4L} a_i$. As a result, we can cover the space of feasible $\boldsymbol{w}_i$'s with $O\left(\left(\frac{L}{\eta}\log M\right)^L L\right)$ number of points. This proves that for a fixed underlying distribution for every $\xi > 0$, with probability at least $\xi > 0$ for any feasible norm-bounded $\boldsymbol{w}$ we have

$$\mathcal{L}_{0-1}^{\mathrm{adv}}(\boldsymbol{w}) - \hat{\mathcal{L}}_\gamma^{\mathrm{adv}}(\boldsymbol{w}) \le \mathcal{O}\left( \sqrt{\frac{B^2 L^2 d\log(Ld)\lambda^2 \left(\prod_{i=1}^L \|\boldsymbol{w}_i\|_\sigma \sum_{i=1}^L \frac{\|\boldsymbol{w}_i\|_2^2}{\|\boldsymbol{w}_i\|_\sigma^2}\right)^2 + L\log\frac{mL\log(M)}{\eta\xi}}{m\gamma^2(\lambda - (1+B)\mathrm{Lip}(\nabla\ell \circ f_{\boldsymbol{w}}))^2}} \right). \quad (107)$$

To apply the result to the network of $n$ nodes, we apply a union bound to have the bound hold simultaneously for the distribution of every node, which proves for every $\xi > 0$ with probability at least $1 - \xi$ the average worst-case loss of the nodes satisfies the following margin-based bound:

$$\mathcal{L}_{0-1}^{\mathrm{adv}}(\boldsymbol{w}) - \hat{\mathcal{L}}_\gamma^{\mathrm{adv}}(\boldsymbol{w}) \le \mathcal{O}\left( \sqrt{\frac{B^2 L^2 d\log(Ld)\lambda^2 \left(\prod_{i=1}^L \|\boldsymbol{w}_i\|_\sigma \sum_{i=1}^L \frac{\|\boldsymbol{w}_i\|_F^2}{\|\boldsymbol{w}_i\|_\sigma^2}\right)^2 + L\log\frac{nmL\log(M)}{\eta\xi}}{m\gamma^2(\lambda - (1+B)\mathrm{Lip}(\nabla f_{\boldsymbol{w}}))^2}} \right). \quad (108)$$

Therefore, the proof is complete.

## Appendix G  Proof of Theorem 4

Define random vector $\mathbf{U} = \Lambda\mathbf{X} + \delta$. According to the definition of optimal transport cost $W_c(P_\mathbf{X}, P_\mathbf{U})$ for quadratic $c(\mathbf{x}, \mathbf{u}) = \frac{1}{2}\|\mathbf{x} - \mathbf{u}\|_2^2$,

$$W_c(P_\mathbf{X}, P_\mathbf{U}) \coloneqq \min_{P_{\mathbf{X},\mathbf{U}}\in\Pi(P_\mathbf{X}, P_\mathbf{U})} \mathbb{E}\left[\frac{1}{2}\|\mathbf{X} - \mathbf{U}\|_2^2\right] \quad (109)$$

where $\Pi(P_{\mathbf{X}}, P_{\mathbf{U}})$ contains any joint distribution $P_{\mathbf{X},\mathbf{U}}$ with marginals $P_{\mathbf{X}}, P_{\mathbf{U}}$. One distribution in $\Pi(P_{\mathbf{X}}, P_{\mathbf{U}})$ is the joint distribution of $(\mathbf{X}, \Lambda\mathbf{X} + \delta)$ implying that

$$
\begin{aligned}
W_c(P_{\mathbf{X}}, P_{\mathbf{U}}) &\le \frac{1}{2}\mathbb{E}\big[\|\mathbf{X} - \Lambda\mathbf{X} - \delta\|_2^2\big] \\
&= \frac{1}{2}\mathbb{E}\big[\|(I-\Lambda)\mathbf{X} - \delta\|_2^2\big] \\
&\overset{(a)}{\le} \mathbb{E}\big[\|(I-\Lambda)\mathbf{X}\|_2^2\big] + \|\delta\|_2^2 \\
&\overset{(b)}{\le} \operatorname{Tr}\big((I-\Lambda)(I-\Lambda)^{\top}\mathbb{E}[\mathbf{X}\mathbf{X}^{\top}]\big) + \|\delta\|_2^2 \\
&\overset{(c)}{\le} \lambda\operatorname{Tr}\big((I-\Lambda)(I-\Lambda)^{\top}\big) + \|\delta\|_2^2 \\
&\overset{(d)}{\le} \lambda\|I-\Lambda\|_F^2 + \|\delta\|_2^2 \\
&\le \max\{\lambda, 1\}\big(\|I-\Lambda\|_F^2 + \|\delta\|_2^2\big).
\end{aligned}
$$

In the above, $(a)$ holds since for every two vectors $\mathbf{u}_1, \mathbf{u}_2$ we have $\|\mathbf{u}_1 + \mathbf{u}_2\|_2^2 = \|\mathbf{u}_1\|_2^2 + \|\mathbf{u}_2\|_2^2 + 2\mathbf{u}_1^{\top}\mathbf{u}_2 \le 2(\|\mathbf{u}_1\|_2^2 + \|\mathbf{u}_2\|_2^2)$. $(b)$ follows from the fact that $\mathbb{E}\big[\|(I-\Lambda)\mathbf{X}\|_2^2\big] = \mathbb{E}\big[\operatorname{Tr}((I-\Lambda)\mathbf{X}\mathbf{X}^{\top}(I-\Lambda)^{\top})\big] = \operatorname{Tr}\big((I-\Lambda)(I-\Lambda)^{\top}\mathbb{E}[\mathbf{X}\mathbf{X}^{\top}]\big)$. $(c)$ holds because of the theorem's assumption implying that $\mathbb{E}[\mathbf{X}\mathbf{X}^{\top}] \preceq \lambda I$. Last, $(d)$ holds because we have $\operatorname{Tr}(AA^{\top}) = \|A\|_F^2$ for every $A$. Therefore, the proof is complete.