[Reviews · NeurIPS 2020]

Review 1

Summary and Contributions: This paper studies the problem of heterogeneity in federated learning. Using a structured model of affine heterogeneity, the authors derive a min-max optimization problem. Borrowing from the adversarial optimization literature, a new algorithm, FedRobust, is introduced in order to have both low communication and computation costs. A first theoretical result demonstrates the convergence of the algorithm to a saddle point. The authors then draw a connection between the Wasserstein distance and the introduced regularisation to prove robustness to Wasserstein distribution shifts of the solution of the problem. Experiments on MNIST and CIFAR 10 with deep neural networks show an improved or better accuracy with respect to other adversarial training methods.

Strengths: - This paper introduces a novel and fruitful connection between adversarial training and federated learning, through a simple structured heterogeneity model. - The authors provide a novel lightweight algorithm which is applicable in the FL setting, with a comprehensive theoretical analysis and compelling experimental results. - This paper opens several avenues for tackling heterogeneity problems. As such, this paper is very relevant for the NeurIPS community.

Weaknesses: - Although the algorithm was designed for large-scale FL, the paper only reports experiments with custom splits of MNIST or CIFAR10 with only 10 nodes. The more realistic LEAF benchmark dataset would have been better in this case. - As the algorithm is motivated by communication and computation costs, a clearer analysis of the advantages of the proposed algorithm over other algorithms in these aspects would have been appreciated.

Correctness: The experiments prove that the proposed algorithm yields a better or same robustness as other algorithms. This is not entirely intuitive, as the proposed algorithm is supposed to be designed to be lightweight, so probably at the expense of robustness. However, I did not understand well the computation and communication advantages over other approaches. In particular, the authors state lines 314-315 that FedRobust is four times faster than distributed PGD: what does it mean? Is it in terms of raw runtime, of local computation time, or communication time?

Clarity: The paper is overall clear and well written; the problem derivation in particular is particularly pleasant to read. However, the readability could be improved for readers not familiar with the adversarial optimization literature. Notably, some standard abbreviations are used without being defined and without providing references: FGSM (l 95), FGM (l 303). PGD is used line 95 but introduced line 300.

Relation to Prior Work: The authors clearly distinguish their work from previous work. However, previous work related to adversarial optimization could be clarified.

Reproducibility: Yes

Additional Feedback: - The proposed algorithm has a noticeable boost in performance over Distributed-PGD/FGM for AlexNet, and a less remarkable one for Resnet and Inception (figs 1 and 2). How do the authors explain these differences? - What is the typical effect of \tau on the performance? - Typo line 97: minimix -> minimax


Review 2

Summary and Contributions: The main contribution of this paper is to develop a federated learning algorithm named FLRA that is robust against affine distribution shifts in users’ samples. FLRA solves a distributed minimax optimization problem using gradient descent ascent. Furthermore, they provide convergence guarantees for minimax objectives that satisfy the Polyak-Lojasiewicz (PL) condition, as well as generalization bounds from empirical distributions of samples. They also perform numerical simulations and compare the performance of the proposed method with other standard federated algorithms.

Strengths: The strength of the paper is in providing a simple yet powerful algorithm that doesn’t require a lot of computation and communication at every iteration. Moreover, it provides an abundance of theoretical guarantees for the performance, generalization, and distributional robustness of the algorithm.

Weaknesses: - The numerical results only compare the test accuracy of the learned model over different distribution shifts and perturbations. While this verifies the robustness of the model and validates the problem formulation, it could be improved by adding figures that represent the computation and communication complexity and training time of the FedRobust, as well as its scalability to the network size. As it is mentioned in the paper, “a typical federated learning consists of a network of hundreds to millions of devices”. However, in the numerical simulations, the number of nodes is selected as n=10. - The convergence guarantee provided in Thm 2 is not too different (albeit a generalization) from earlier results on distributed minimax optimization. - The generalization guarantees follow earlier work; there is not substantial innovation in the result presented in Thm. 3.

Correctness: Yes

Clarity: The paper is well written. However, there are some minor issues explained below: • How is the parameter \lambda chosen given the bounds on the affine distribution shifts? • Assumption 2 is stated but does not seem to be used in any of the theorems.

Relation to Prior Work: Yes

Reproducibility: Yes

Additional Feedback: ***************** Comments after the rebuttal *************** After reading the paper again and the reviews, I believe the paper has several limitations: - the affine robustness is limited, even if slightly more general than some earlier models. - the convergence results are similar to work on local SGD. There one proves convergence to a global min (in the convex case) or a local min (in a non-convex case); here they show convergence to a saddle-point for the min-max problem. - the generalization guarantees are based on earlier work. In short, IMO, the paper puts nicely a bunch of things together for a limited robustness model and does not have a key innovation.


Review 3

Summary and Contributions: EDIT: After reviewing the author feedback, I have revised my score to a 7. The authors provided useful details in the feedback regarding their empirical setup, differences between FedRobust and PGD, and the theoretical novelty in the paper. Thus, my revised score is in anticipation of this being worked into the final draft of the paper. -------------------- The paper proposes a variant of federated averaging (FedRobust) designed for settings where the client data distributions vary by some affine shift. The authors analyze a few important facets of this problem setup and their algorithm. First, they show that under a 1- and 2-sided PL condition, the algorithm converges in an asymptotically similar manner to non-federated robust optimization. Second, they analyze generalization properties of their robust loss function via spectral regularization. Finally, they give a kind of framing of their affine shift problem in terms of distributional robustness. These theoretical results are complemented by experiments showcasing the utility of FedRobust in learning models robust to affine shifts. Interestingly enough, the authors also show that their method helps guard against Projected Gradient Descent (PGD) attacks common in the adversarial robustness literature.

Strengths: An important facet of this work is that the authors really attempt to study robustness in a federated setting, not simply in a setting with some amount of distributed computation. In particular, the authors are cognizant of systems-level factors in federated learning, such as limited communication costs. The authors also develop a quite sophisticated (and potentially more generally useful) theoretical framework for understanding minimax problems in federated learning. In particular, the authors are able to show that doing the maximization step on the clients (not on the server) is sufficient, and that the server can perform its standard FedAvg step. While the work builds heavily on theory from non-federated works (such as work on spectrally normalized generalization bounds for neural networks (eg. Bartlett et al., 2017, or recent work by Yang et al. on minimax problems), the extra steps needed to make these relevant to FL is novel and important.

Weaknesses: One of the weaknesses in the empirical section is, I believe, an opportunity to make the paper better. The authors observe that FedRobust is seemingly better at ensuring adversarial robustness to attacks than the de facto standard methods (PGD, FGSM). The reason for this is not clear, especially since these methods were designed to defend against PGD attacks. Such an empirical result I think necessitates a detailed explanation, or at least hypothesis, as to what is happening. While I believe that one can view PGD defenses as a kind of limited version of FedRobust that only add a bias term (they do not scale), the fact that PGD does so more local updates and yet does not do better is somewhat confusing. A few other aspects of the empirical study are slightly strange. For example, the authors found similar accuracy of PGD and FGM when defending against PGD attacks. This contradicts some of the empirical results in (Madry et al., 2019), the paper that designed the PGD defense. This could mean that the attack regime used is relatively weak. In this case, it would be beneficial to showcase more drastic attacks, to see if FedRobust continues to perform well against them. While some such figures are given in Figure 4 in the appendix, these also find that oftentimes FGM is worse than FedAvg, which again necessitates an explanation. Finally, not all the empirical details are fully described (at least, as best as I can tell). When describing the FGM and PGD defense methods evaluated, the authors do not describe the step-size used for these defenses. This learning rate can be critical to the performance of these methods (see Madry et al., 2017).

Correctness: I believe that by and large, the theoretical results are entirely correct. The authors do a commendable job in the appendix of building up a coherent and detailed theory of federated minimax optimization. However, this theory is not discussed at all in the main body of the work. Some clue as to what techniques are used to prove this result may be useful, and could inspire people to use similar techniques more generally. Theorem 1 is also something that could be improved for the sake of correctness. In particular, the assumptions of the theorem require 4 separate conditions on the 2 learning rates used in FedRobust. It is not clear from an inspection of these conditions whether extra assumptions are needed for there to exist some \eta_1 and \eta_2 satisfying these conditions. A simplification of these conditions, even at the expense of slightly worse constants, would go a long way to making the theorem more digestible. For example, non-federated minimax optimization results in Yang et al., 2020 (which are cited in this paper as being roughly comparable) generally utilize simpler learning rate conditions that are more obviously satisfiable. The experiments look correct as well, save for the behavior discussed above regarding the performance of FedRobust, PGD, FGM, and FedAvg on PGD attacks. Explanation as to what's happening here would be useful to assure the reader of the validity of the results.

Clarity: The paper is generally well-written. The authors do a very good job of discussing federated learning, robust optimization, and the interplay between the two. They also spend a lot of time in helping the reader understand the exact robust optimization setting being considered, which can be immensely helpful to the layman trying to understand the paper. The authors also do a good job of discussing many separate important theoretical areas, including minimax optimization, generalization, and distributional robustness. However, I wish that there was a bit more of a coherent flow as to why all three of these aspects are considered. Is there a reason that we should be concerned with all three simultaneously? As it stands, the theorems in the work read a little like a list of results, rather than contributing to an overarching message. I also believe that more discussion could be made of the empirical results (as discussed above), and their implications. On a related note, it would be useful for the authors to address the question of whether guarding against affine shifts in distributions is useful for ensuring more general forms of robustness. This attitude is advocated in the broader impact statement, with assertions that FedRobust can help all kinds of robust federated learning, but without much discussion as to why the authors believe this. To allow for this extra explanation, I also believe the authors can (and should) pare down the introduction sections. The authors spend almost 3 pages just giving introduction. While these sections are useful, they are relatively wordy, and can definitely be reduced. For example, the authors spend nearly an entire page discussing their own contributions in Section 1. By contrast, there is less than half a page on related work, and virtually no discussion of how sections 4.1, 4.2, and 4.3 tie together. While these are examples that stood out to me, I would encourage the authors to think about the message they are trying to send, and how best to do so.

Relation to Prior Work: While I think the authors do a reasonable job of covering related work, I think that they attribute a few too many things as being novel in their approach compared to past work. For instance, one of the benefits of FedRobust they highlight in L182-188 is its communication efficiency stemming from doing synchronization of models only every \tau rounds. This is already the de facto standard for federated learning algorithms, and effectively has been since (McMahan et al., 2017) was published. The fact that FedRobust is essentially a modification of FedAvg that utilizes client gradient ascent steps is not discussed explicitly. While the authors cite basically all the relevant work to their own, some of the discussions lack a nuance of what new contributions are made in this work. For example, Theorem 3 presents a generalization bound based on the well-known technique of spectral normalization (eg. Bartlett et al., 2017). It is unclear what specific improvements were made in this work over other such bounds. In short, I would suggest that the authors spend more time actually differentiating their theoretical results, rather than trying to disambiguate their algorithm from other algorithms, as I believe that the theoretical contributions in this paper are its strongest aspect.

Reproducibility: No

Additional Feedback: All relevant suggestions have been made above. However, in the interest of summarizing, here are my suggestions for improving the paper. 1. Include an expanded discussion of the empirical results. This includes more details about the experimental setup (eg. what learning rates are used for the PGD and FGM defenses), as well as discussing the relative efficacy of these methods. In particular, why FedRobust seems to do comparably to PGD defenses against PGD attacks, and in what settings we might start to see substantial divergence of these methods. 2. Better discussion of the theoretical results in this paper. This includes more comprehensible conditions on the learning rate in Theorem 1, as well as more discussion of the actual theoretical advances made by this paper. I believe that the theory in this paper could have significant uses in many other works, but currently it is difficult to extract the specific techniques used in the appendix. 3. The writing of this paper could be improved by giving a more concise discussion of this paper's contributions, and using the extra page space to discuss a more coherent message. In particular, it would be useful for the reader to understand why Theorems 1-4 are useful in tandem, not just as separate results, and how they are reflected in the empirical results.


Review 4

Summary and Contributions: The paper considers the problem of Federated learning in which the different devices / nodes may have different distributions of user data (e.g consider pictures taken from different smartphones by different users, etc). It is known that standard FL methods have poor accuracy in such senarios. The authors use ideas from DRO (distributionally robust optimization) to propose a principled algorithm which mitigates this heterogeneity issue. The ambiguity sets used in the proposed framework are Wasserstein balls containing transformed versions of a certain (unknown) underlying base distribution (e.g different smartphone cameras have different characteristics). The result is a tractable online algorithm minimax algorithm. Extensive results are shown on real-word datasets and the authors show that their proposed algorithm outperforms comptetitors.

Strengths: - The paper is extremely well-written and easy to follow. - The algorithms motivated, presented, and duely discussed. - Explicit convergence rates are also given, and match the classical scenario in the homogeneous limit when the size of the ambiguity set is shrunk to zero. - Proofs of all theoretical results presented in the paper are provided in the supplemental. - The experimental section is strong. Benchmarks include multiple datasets and genuine competing methods. Empirical results are convincing.

Weaknesses: - The affine transformation drift model is easy to analyse but I wonder to what extend this can be used in practical FL scenarios.

Correctness: - Everything seems correct. Theorems are clearly-stated and discussed. Complete proofs are provided in supplemental. - Extensive experimental results are presented and discussed. Benchmarks include reasonable competitors (FedAvg, PGD, FGM, etc.).

Clarity: The paper is extremely well-written and easy to follow.

Relation to Prior Work: Yes, the authors clearly situate their contributions within the vast literature on the subject.

Reproducibility: Yes

Additional Feedback: - It would be interesting to study what happens in the nonparametric regime where the contraints are replaced with something as general as W(P_X, P^i_X) <= eps. The worst-case adversarial distribution of data at the device i, namely P^i_X will correspond to (non-affinely) moving the data around. In certain cases, in certain cases, would still lead to a tractable program. - What happens in the experiments if we drop the Gamma matrices and only consider translations ? I've expect this to already lead to an improvement over the SOTA. The problem would also be considerably easier than with the Gamma matrices involved. Another simplification would be to only consider diagonal Gamma matrices.

[Author Response · NeurIPS 2020]

We thank the reviewers for their careful consideration and their feedback. We provide our responses below.

**Response regarding numerical results.** -*"Adversarial robustness gained by `FedRobust` compared to distributed PGD and FGM methods:"* **(R1, R3)** As noted by the reviewers, `FedRobust` achieved a similar (or superior) adversarial robustness to the standard PGD training. This observation can be explained by analyzing the generalization properties of these algorithms. We note that `FedRobust`'s improved robustness was obtained over the **test** samples. On the other hand, PGD consistently outperformed `FedRobust` on the **training** samples, achieving a near perfect training accuracy. However, `FedRobust` generalized better to the test samples and could overall outperform PGD on the test set. We will include the training performance scores in the final version for clarification. Also, the similar performance of FGM and PGD can be explained via the random Gaussian perturbations used for simulating the heterogeneity across clients and the results of Wong et al. (ICLR 2020) indicating FGM initialized at random perturbations performs as well as PGD.

-*"Effect of network size $n$ and # of local updates $\tau$; other datasets:"* **(R1, R2)** We have performed several new experiments to study the effect of larger $n$ and $\tau$ and will add the new results. Here, we report some preliminary results for $n = 100$ (top) and $\tau = 5$ (bottom). Both results indicate that `FedRobust` still offers a significant robustness gain over `FedAvg`. We will also conduct experiments using the suggested LEAF framework (FEMNIST dataset).

-*"Computation and communication times in speed comparisons:"* **(R1, R2)** The time comparison made in the main body is in terms of the computation time for the same number of training iterations. We note that in our experiments the methods will share the same communication time as they have been trained for the same $10,000$ iterations (and rounds).

-*"Step-size of PGD and FGM:"* **(R3)** For PGD training, we used the standard rule of thumb to choose step-size $\frac{2}{k}\epsilon_{\text{pgd}}$ for each of $k = 10$ PGD steps. For FGM training, the effective step-size is the same as $\epsilon_{\text{fgm}}$, since FGM normalizes the single-step perturbation.

**Response regarding theoretical results.** -*"Summary of contributions, technical advances and Theorems 1-4 in tandem:"* **(R3)** We first consider a heterogeneity model where the data distribution at each node is an affine transformation of a mother distribution. Using this model, we formulate a robust federated optimization problem in eq. (3). To solve this minimax problem, we propose a communication-computation efficient optimization algorithm (`FedRobust`) and show its convergence (Theorems 1, 2). This paper is the first work that integrates proof techniques from local SGD, minimax optimization, federated optimization, and provides provable robust federated methods. Other existing results in distributed minimax optimization and non-robust federated learning can be retrieved as special cases of our results. Then in Theorem 3, we ensure that when a new client with unseen data joins the federated network, the model learned by solving (3) is properly generalized. Finally, in Theorem 4 we connect our proposed minimax formulation (3) to distributionally robust optimization by showing that it indeed optimizes a lower-bound on the distributionally robust problem with Wasserstein cost in (7).

-*"Theorem 2 vs. prior results in distributed minimax optimization, effect of $\tau$:"* **(R1, R2)** Theorems 1, 2 characterize the convergence rates of `FedRobust` in which, each of the clients runs $\tau$ local updates in each round. General distributed minimax optimization algorithms can be viewed as special cases for $\tau = 1$. The effect of running more than one local update ($\tau > 1$) in the convergence rates are demonstrated in both theorems by the terms containing $(\tau - 1)$. Analysing the effect of $\tau > 1$ is indeed a technical challenge in our convergence analysis **(R2)**. At a high-level, $\tau$ controls the computation-communication trade-off as larger $\tau$ implies less communication at the expense of more computation **(R1)**.

-*"Technical novelty of Theorem 3:"* **(R2, R3)** We note that the distribution shift considered in this paper is device-dependent, i.e. all the samples stored at node $i$ undergo the same transformation $\Lambda^i \mathbf{x} + \delta^i$. This is unlike the prior works in adversarial training such as Farnia et al. (2018), where *each* data sample is affected by a different transformation. Moreover, the affine shift considered in our paper is specified with two variable $\Lambda, \delta$, generalizing over the prior works considering only $\delta$. These two challenges distinguish Theorem 3 **(R2)**. We also note that our proof of Theorem 3 uses the PAC-Bayes framework (McAllester, 1999), while Bartlett et al. (2017) analyzes the Rademacher complexity **(R3)**.

-*"Discussion on learning rates:"* **(R3)** The conditions on $\eta_1, \eta_2$ in Theorem 1 can be rewritten as linear constraints and are always feasible. Rewriting the last condition as linear in $\eta_1, \eta_2$: $\eta_1 \hat{L} + 40(3\eta_1 L_1^2 + \eta_2 L_{21}^2)(\tau - 1)\mu_1^{-1}(1 - \frac{\mu_1}{8\sqrt{2}L_1})^{-1} \leq 1$. E.g.: $\eta_1 = c_1 \ln(T)/T$, $\eta_2 = c_2 \ln(T)/T$ where $c_2 = [160\kappa n L]^{-1}$ and $c_1 = \min\{[(482\kappa + 6)\tau L]^{-1}, [1440\kappa^3 n L]^{-1}\}$.

-*"Practicality of the affine model in FL; nonparametric regime; effect of matrices $\Lambda$"* **(R4)** The affine model considered in this paper is particularly practical for image classification tasks in FL as also elaborated in the introduction, where each camera's imperfections affect its pictures (Robey et al., 2020). While this model provides significant robustness compared to additive-only perturbation models (i.e. $\Lambda = I$), it lays out potential new directions to study more complicated (non-affine) models such as neural network transformations. The nonparametric regime is another interesting generalization of this work, however in this case, nodes might need to solve a maximization problem at each iteration which can be problematic due to limited computation in federated settings. Diagonal $\Lambda$ is also another interesting special case, however it may also fall short in capturing different filtering functions, e.g. rotation of images.

[Meta-Review · NeurIPS 2020]

The paper studies robust federating averaging under affine shifts. All the reviewers agree that the paper has novel and interesting result. For camera ready, please take reviewers' feedback into account. In particular, a key weakness of the work is it's restriction to the affine setting. Comments on why the setting is interesting and if/how the result can be extended to more general setting would be useful.